# Denitrification and associated nitrous oxide and carbon dioxide emissions from the Amazonian wetlands

Jérémy Guilhen[1,2], Ahmad Al Bitar[2], Sabine Sauvage[1], Marie Parrens[2,3], Jean-Michel Martinez[4], Gwenael Abril[5,6], Patricia Moreira-Turcq[7], and José-Miguel Sánchez-Pérez[1]

[1]Laboratoire Ecologie Fonctionnelle et Environnement (EcoLab), Institut national polytechnique de Toulouse (INPT), CNRS, Université de Toulouse (UPS), France
[2]Centre d'Observation de la Bioshpère (CESBIO), CNES, Université de Toulouse (UPS), France
[3]Dynafor, Université de Toulouse, INRAE, INPT, INP-PURPAN, Castanet-Tolosan, France
[4]Géosciences Environnement Toulouse (GET), IRD/CNRS, Université Toulouse (UPS), France
[5]Biologie des Organismes et Ecosystèmes Aquatiques (BOREA), Muséum National d'Histoire Naturelle, Paris, France
[6]Programa de Geoquímica, Universidade Federal Fluminense, Outeiro São João Batista, Niterói, RJ, Brazil
[7]IRD (Institut de Recherche pour le Développement), GET (Géosciences Environnement Toulouse), UMR 5563, Lima, Peru

**Correspondence:** J. Guilhen (jeremy.guilhen@gmail.com) and S. Sauvage (sabine.sauvage@univ-tlse3.fr)

**Abstract.** In this paper, we quantify the $CO_2$ and $N_2O$ emissions from denitrification over the Amazonian wetlands. The study concerns the entire Amazonian wetland ecosystem with a specific focus on three floodplain (FP) locations: the Branco FP, the Madeira FP and the FP alongside the Amazon River. We adapted a simple denitrification model to the case of tropical wetlands and forced it by open water surface extent products from the Soil Moisture and Ocean Salinity (SMOS) satellite. A priori model parameters were provided by in situ observations and gauging stations from the HyBAm observatory. Our results show that the denitrification and the trace gas emissions present a strong cyclic pattern linked to the inundation processes that can be divided into three distinct phases: activation - stabilization - deactivation. We quantify the average yearly denitrification and associated emissions of $CO_2$ and $N_2O$ over the entire watershed at 17.8 kgN/ha/yr, 0.37 gC-$CO_2$/m$^2$/yr and 0.18 gN-$N_2O$/m$^2$/yr respectively for the period 2011-2015. When compared to local observations, it was found that the $CO_2$ emissions accounted for $0.01\%$ of the integrated ecosystem, which emphasizes the fact that minor changes to the land cover may induce strong impacts to the Amazonian carbon budget. Our results are consistent with the state of the art of global nitrogen models with a positive bias of 28%. When compared to other wetlands in different pedo-climatic environments we found that the Amazonian wetlands have similar emissions of $N_2O$ with the Congo tropical wetlands and lower emissions than the temperate and tropical anthropogenic wetlands of the Garonne river (France), the Rhine river (Europe), and south-eastern Asia rice paddies. In summary our paper shows that a data-model-based approach can be successfully applied to quantify $N_2O$ and $CO_2$ fluxes associated with denitrification over the Amazon basin. In the future, the use of higher resolution remote sensing product from sensor fusion or new sensors like the Surface Water Ocean Topography Mission (SWOT) mission will permit the transposition of the approach to other large scale watersheds in tropical environment.

# 1 Introduction

Inland waters play a crucial role in the carbon and nitrogen cycle. In particular, wetlands sequester atmospheric and fluvial carbon (Abril and Borges, 2018). This phenomenon is intimately linked to nitrous oxide ($N_2O$) (Wu et al., 2009) and carbon dioxide ($CO_2$) emissions to the atmosphere (Borges et al., 2015). In wetlands, during inundation periods denitrification pro-

cesses nitrates ($NO_3^-$) into atmospheric dinitrogen ($N_2$). These processes are controlled by biogeochemical reactions linked to microorganisms activity and pedoclimatic conditions (soil characteristics, nutrients availability and water content). Moreover, the alternations between dry and wet periods in wetlands promote carbon and nitrogen mineralization and denitrification in soils (Koschorreck and Darwich, 2003). Our understanding and capacity to quantify the mechanisms involved in $N_2O$ and $CO_2$ emissions over wetlands are limited and lead to uncertainties in estimating them at large scales.

During the last decade, process-based models have become key tools in estimating carbon and nitrogen budgets in the context of global multi-source changes. Recent studies presenting a review of existing models capable of quantifying $N_2O$ and $CO_2$ fluxes over continental ecosystems (Tian et al., 2018; Lauerwald et al., 2017) show that they are mainly used to characterize the part of greenhouse gases (GHGs) emissions due to natural and anthropogenic/agricultural activities at different spatiotemporal scales. The estimation of $N_2O$ emissions from natural sources are still subject to large uncertainties (Ciais and Coauthors.,

2013) while $N_2O$ emissions from anthropogenic activities are under investigation. Assessing $N_2O$ budget for wetlands at large scale currently constitutes a knowledge gap. In terms of denitrification, the relatively sparse and shot-term observations limit our capability to estimate the carbon and nitrogen recycling in terrestrial ecosystems, especially over wetlands. Since in situ measurements constitute the main source of data, few studies assess $N_2O$ and $CO_2$ emissions from denitrification at large scale and are usually limited to field scale or small scale watersheds (Russell et al., 2019; Johnson et al., 2019; Korol et al., 2019).

In the case of the Amazon basin, the total amount of $CO_2$ emission reaches 0.3 PgC/yr for both natural and agricultural sources. Scofield et al. (2016) pointed out that over the Amazonian wetlands disproportionally high $CO_2$ out-gassing may be explained by the abundant amount of podzols for the Negro Basin. Podzols slow the organic matter decomposition and increase the leaching of humus. Over the Amazon basin, floodplain soils are mainly Gleysols (Legros, 2007) which are characterized by high microbiological activity. $CO_2$ emissions from the river are mainly due to organic matter respiration as well as exports from the

wetland system. In wetlands, root respiration and microbial activities are a major source of $CO_2$ emissions (Abril et al., 2014). Ultimately $CO_2$ outgassed from the Amazon River is about $145 \pm 40$ TgC/yr (de Fatima F. L. Rasera et al., 2008) and tops at 470 TgC/yr when extrapolated to the whole basin (Richey et al., 2002). In regards to the carbon budget, some studies show that the Amazon basin is more or less in balance and even acts as a small sink of carbon at the amount of 1GtC/yr (Lloyd et al., 2007).

Remote sensing has emerged as a major tool for GHGs quantification, either via assimilation into physically-based models (Engelen et al., 2009) or as a direct observation (Bréon and Ciais, 2010). For wetlands, the monitoring of water extents is crucial for the denitrification processes. Water surface monitoring has been done with a variety of spectral bands (Martinez and Le Toan, 2007; Pekel et al., 2016; Birkett et al., 2002) in active and passive remote sensing. Recently L-Band microwave remote sensing showed advanced capabilities to monitor water surfaces in tropical environment because of all-weather capa-

bilities, providing soil signal under vegetation (Parrens et al., 2017).

This study aims to deliver an enhanced understanding and quantification of the denitrification process over the Amazonian wetlands with their associated fluxes of $N_2O$ and $CO_2$ using modelling and microwave remote sensing. We constrained and adapted a denitrification process-based set of equations by L-Band microwave water surface extents from the Soil Moisture

5   and Ocean Salinity (SMOS) satellite and a priori in situ information. The specific objectives of the study are to highlight the main key factors controlling the denitrification and to identify the hot spots and hot moments of denitrification over wetlands. A hot spot represents an area that shows disproportionately high reaction rates relative to the surrounding and a hot moment corresponds to a short period of time with disproportionately high reaction rates relative to longer intervening time periods (McClain et al., 2003).

## 2   Materials and methods

### 2.1   Study area

The Amazon basin (Fig.1) is the world's largest drainage basin with an area of $5.50 \times 10^6$ km$^2$ and an average water discharge of $208\,000$ m$^3$ s$^{-1}$ (Callode et al., 2010) representing $20\%$ of all surface freshwaters transported to the ocean. The watershed

spans across Bolivia, Colombia, Ecuador, French Guiana, Peru, Suriname, and Guyana and $68\%$ of the basin pertains to Brazil.

Devol et al. (1995) described the hydrology of the main stream as the aggregation of the water originating from Andean regions, from the main tributaries and from "local sources" corresponding to smaller streams draining local lowlands. The contribution of each water body differs in time. For example from November to May the contribution of Andean waters reaches $60\%$ and

declines during the dry season to $30\%$. Wetlands are essential in the watershed functioning: $30\%$ of the Amazon discharge has once passed through the floodplain distributed along a 2010 km reach between São Paulo de Olivença and Òbidos (Richey et al., 1990).

The Amazon basin contains several floodplains (FP). Here we consider three main floodplains: the Branco FP in the northern part, the Madeira FP in the southern part and the floodplain between Odidos and Manaus which is called Obidos-Manaus

floodplain (in the following O-M FP). The O-M FP covers an area of $2.50 \times 10^5$ km$^2$ whereas the Madeira FP covers $3.70 \times 10^5$ km$^2$. The Branco FP is the widest of the three floodplains with a covered area of $6.70 \times 10^5$ km$^2$.

### 2.2   Materials

#### 2.2.1   In situ data from the HyBAm observatory

In situ data were obtained from the Hidro-geoquímica da Bacia Amazônica (HyBAm) long-term monitoring network that

maintains, in collaboration with the national stakeholders and local universities, 13 gauging stations in the Amazon catchment basin since 2003. For the Brazilian part of the basin, a network of eight local stations is maintained by the French Research

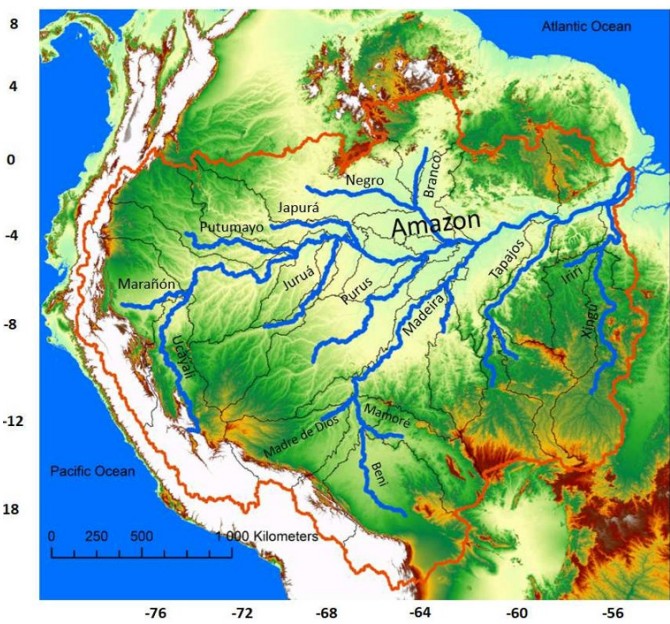

**Figure 1.** The Amazon river basin and its main tributaries mapped over the SRTM (Shuttle Radar Topography Mission - 500 m) digital elevation model.

Institute for Development (IRD) and the Amazonas Federal University (UFAM). Geochemical, sedimentary and hydrological data are available freely at www.so-hybam.org for each gauging station. River discharge records are available daily while geochemical data, including Dissolved Organic Carbon (DOC), are available monthly. In our study, we extracted both the daily river discharges and the monthly DOC concentrations.

### 2.2.2 Water surface extents from L-Band microwave

The Soil WAter Fraction (SWAF) retrieved from L-Band microwave is used to determine the open water surfaces (Parrens et al., 2017). SWAF is obtained using a contextual model to the SMOS angle binned brightness temperatures (MIRCLF3TA) data (Al Bitar et al., 2017). SMOS was launched in November 2009 by the European Space Agency (ESA) and is the first satellite dedicated to map soil moisture. SMOS is a passive microwave 2-D interferometric radiometer operating in L-band (1.413 GHz, 21 cm wavelength) (Kerr et al., 2010). SMOS orbits at a 757 km altitude and provides Brightness Temperature (TB) emitted from the Earth over a range of incidence angles (0° to 55°) with a spatial resolution of 35 to 50 km. Parrens et al. (2017) showed the capability of SMOS to retrieve the water fraction under dense forests over the Amazon basin. One of the main upsides of SMOS is its sensitivity to soil signal under vegetation in all-weather conditions thanks to the L-Band frequency. The SWAF data were averaged each month over the sampling period (2011-2015) within the Amazon basin. The SMOS satellite observes the Earth surface at full polarization (Horizontal - H, Vertical - V and cross-polarization - HV) at multi incidence angles. In this paper, the SWAF product was generated from the SMOS TB data at 32.5° and V-polarization.

Fig.2 outlines the common hydrological patterns observed in the Amazon basin as well as the dynamic of the inundations for the different floodplains. The contrasted seasonal peaks in flooded areas between the Northern and Southern floodplains are well depicted.

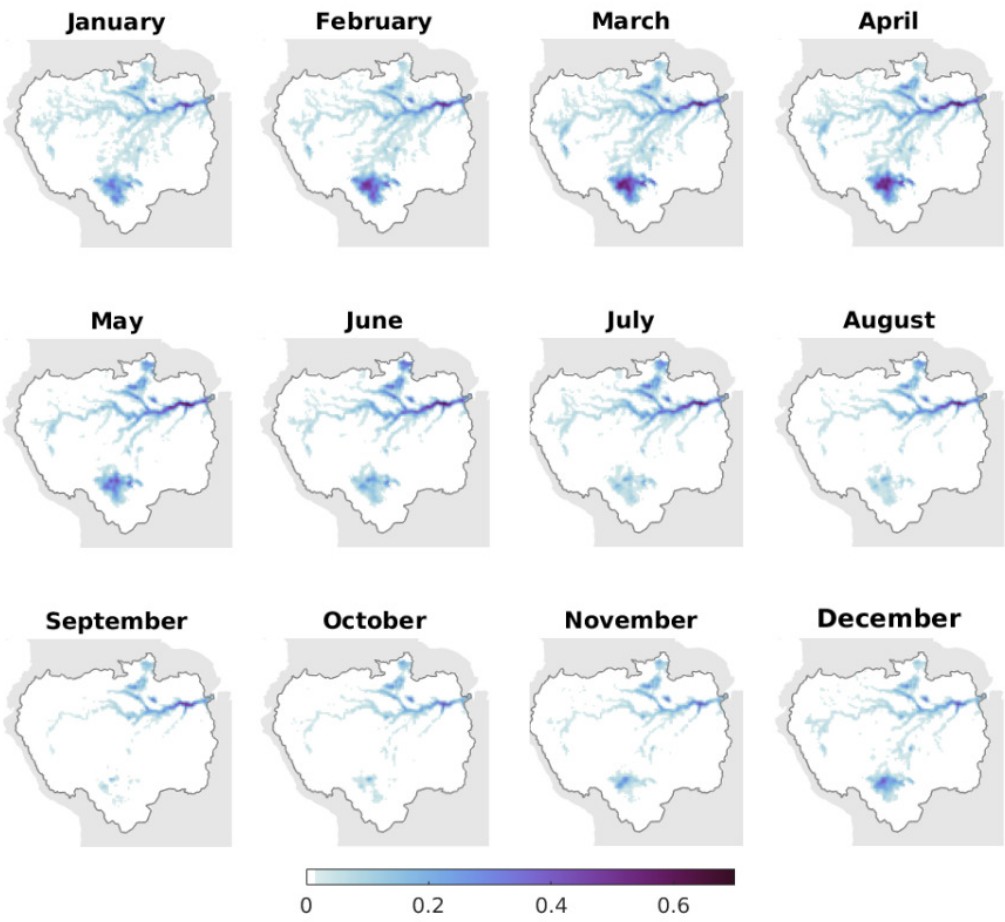

**Figure 2.** Monthly averages from 2011 to 2015 of the SWAF surface water fractions over the Amazon basin based on Vertical polarization Brightness Temperatures (TB V) at 32.5° incidence angle acquired by the SMOS satellite.

## 2.3 Methods

### 2.3.1 Assessing denitrification and emissions

In this study, we modified the denitrification rate proposed by Peyrard et al. (2010) to fit tropical wetland conditions. Denitrification is the consumption of DOC, Particulate Organic Carbon (POC) and nitrate ($NO_3^-$) in the soil. This process is limited by oxygen ($O_2$) and ammonium ($NH_4^+$) availability. Denitrification occurs during flood events when the soil has low $O_2$ con-

centrations, thus $O_2$ concentration is not a limiting factor (Dodla et al., 2008). Furthermore, as there is only one long flood pulse in the Amazon watershed, we consider that all the $NH_4^+$ is processed into $NO_3^-$ between two consecutive floods. We also consider that $NH_4^+$ is not a limiting factor. The fact that $NO_3^-$ stocks are reconstituted by nitrification under aerobic conditions, e.g when soils are no longer flooded, is a reasonable assumption in the case of the Amazon basin and more particularly for the wetland parts as shown by Sánchez-Pérez and Trémolières (2003) on the upper Rhine floodplain. In another work, on the groundwater of the alluvial floodplain of the Garonne river, Iribar et al. (2015) showed that denitrification is the main process that produces $N_2$ and quantified the Nosz involved in heterotrophic denitrification. Besides, many studies consider denitrification as a combined consumption of $NO_3^-$ and carbon (Scofield et al., 2016; Dodla et al., 2008; Goldman et al., 2017). Taking into consideration the above statements, the denitrification rate is expressed as:

$$R_{\mathrm{NO_3}} = -0.8 \cdot \alpha \cdot (\rho \cdot \frac{1-\phi}{\phi} \cdot k_{POC} \cdot [POC] \cdot \frac{10^6}{M_C} + k_{DOC} \cdot [DOC]) \cdot \frac{[\mathrm{NO_3^-}]}{[k_{\mathrm{NO_3}} + \mathrm{NO_3^-}]} \qquad (1)$$

where $R_{\mathrm{NO_3}}$ is the denitrification rate in $\mu mol\,L^{-1}\,d^{-1}$, $0.8 \cdot \alpha$ represents the stoichiometric proportion of $NO_3^-$ consumed in denitrification compared to the organic matter used with $alpha = 5$ as mentioned in Peyrard et al. (2010) , $\rho$ is the dry sediment density $kg\,dm^{-3}$, $\phi$ is the sediment porosity, $k_{POC}$ is mineralization rate constant of POC ($d^{-1}$), $POC$ refers to the POC in the soil and the aquifer sediment (%), $M_C$ is the carbon molar mass $g\,mol^{-1}$, $DOC$ refers to the DOC in the aquifer water $\mu mol\,L^{-1}$, $k_{DOC}$ is the mineralization rate constant of DOC ($d^{-1}$), $k_{\mathrm{NO_3}}$ is the half-saturation for $NO_3^-$ limitation in $\mu mol\,L^{-1}$ and $NO_3^-$ is the nitrate concentration in the aquifer in $\mu mol\,L^{-1}$.

The estimation of $CO_2$ emissions is based on the denitrification equation where gaseous $CO_2$ is formed. We consider that neither $NO_3^-$ nor organic matter are limiting factors for the reaction which is considered total (Eq. 2) (de Freitas et al., 2001). Abril and Frankignoulle (2001) showed that denitrification tends to raise the alkalinity. In order to take into account this phenomenon, the formation of $HCO_3^-$ from dissolved $CO_2$ (Eq. 3) was coupled to the denitrification (Eq. 2).

$$4\,NO_3^- + 5\,CH_2O + 4\,H^+ \longrightarrow 2\,N_2 + 5\,CO_2 + 7\,H_2O \qquad (2)$$

$$CO_2 + H_2O \longrightarrow HCO_3^- + H^+ \qquad (3)$$

Overall, in this study, denitrification was modelled using:

$$4\,NO_3^- + 5\,CH_2O \longrightarrow 2\,N_2 + CO_2 + 4\,HCO_3^- + 3\,H_2O \qquad (4)$$

The equation of the chemical reaction of denitrification (Eq. 4) is used to determine the generated amount of $CO_2$ by relating it to the amount of $NO_3^-$ denitrified. Finally, $N_2O$ production is indirectly estimated as a result of $N_2$ formation. Production of $N_2O$ from $N_2$ during denitrification commonly ranges from a factor of 0.05 to 0.2 (Pérez et al., 2000). Nevertheless, with no precise field measurements an average $N_2O$ / $N_2$ ratio of 0.1 (Weier et al., 1992) was applied in the study.

### 2.3.2 Parametrization of dissolved/particulate organic carbon and nitrate concentrations

The model's parameters for the denitrification are taken from reference studies and in situ measurements. The sediment porosity $\phi$ was set to 25%. It is computed based on the soil texture from the Food and Agricultural Organization (FAO) database at 11 km resolution. The porosity is averaged over the computation nodes (25 km x 25 km) using a bilinear interpolation. $k_{POC}$, $k_{DOC}$ and $k_{NO_3}$ were set to $1.6 \times 10^{-7}$ $d^{-1}$, $8.0 \times 10^{-3}$ $d^{-1}$ and 30 $\mu mol\,L^{-1}$ respectively. They are adapted from (Sun et al., 2017) who performed a study of denitrification over the Garonne catchment (temperate anthropogenic watershed). To our knowledge, these parameters were never measured over the Amazon basin and the values we used are the only published estimates that we have. According to the studies performed by Moreira-Turcq et al. (2013), the POC concentration was considered constant over the whole watershed and for the entire period of the simulation (2011 – 2015) to 10%.

The daily discharge was extracted from the gauging stations used in the study (Fig. 3) from the HyBAm database (1983 – 2012). For each station, we calculated the mean monthly discharge from the daily observations. In terms of discharge, the marked seasonality of the Amazonian streams was demonstrated by prior studies (Paiva et al., 2013). For the DOC concentrations, we extracted the monthly measurements for the same stations over the same period. As the SWAF's period (2011 – 2015) and the DOC measurements are not concomitant, we calculated a mean average monthly DOC concentration for each station. When the information of DOC concentration was not available, our dataset was gap-filled using a linear relationship between DOC concentration and discharge (Ludwig et al., 1996), based on the discharge marked seasonality of the Amazonian streams. Finally, we extended the calculated values to the associated main sub-basin of the gauging station.

NO$_3^-$ concentrations were calculated for every type of soil given by the FAO's classification in the upper 30 cm layer (Fig. 3). Batjes and Dijkshoorn (1999) drew a complete description of the total nitrogen content of the soils of the Amazon region. Evaluating NO$_3^-$ in the upper layer of the soils was executed adapting the mineralization rate which is based on the average temperature of the region and the proportion of both clay and limestone. For the most biologically active soils, as gleysols and fluvisols, the mineralization rate was set up to 7% of the organic nitrogen amount, which is the maximum observed value in the region. On the contrary, regosols are biologically less active soils with mineralization rates hardly reaching 2% (Legros, 2007; Sumner, 1999). Finally, we determined the NO$_3^-$ concentrations by combining the NO$_3^-$ content in each type of soil with the water storage capacity for each type of soil, retrieved from the FAO soil database. NO$_3^-$ concentrations were considered constant over the period. On the one hand, as the Amazon is one of the most active regions of the world (Legros, 2007) in terms of microbial soil dynamic, during non-flooding periods, mineralization of nitrogen was sufficient to compensate NO$_3^-$ lose by plant assimilation and leaching. On the other hand, Sánchez-Perez et al. (1999) showed that when denitrification is active during flood events, the NO$_3^-$ pool of wetlands is provided and sustained by NO$_3^-$ content coming from streams, in the case of the forested Rhine floodplain.

### 2.3.3 Denitrification computation

The methodology focuses on modelling the denitrification process that occurs in the first 30 cm of water-saturated soils in wetlands. Thereby, only the NO$_3^-$ included in that layer were considered undergoing denitrification. NO$_3^-$ brought by streams

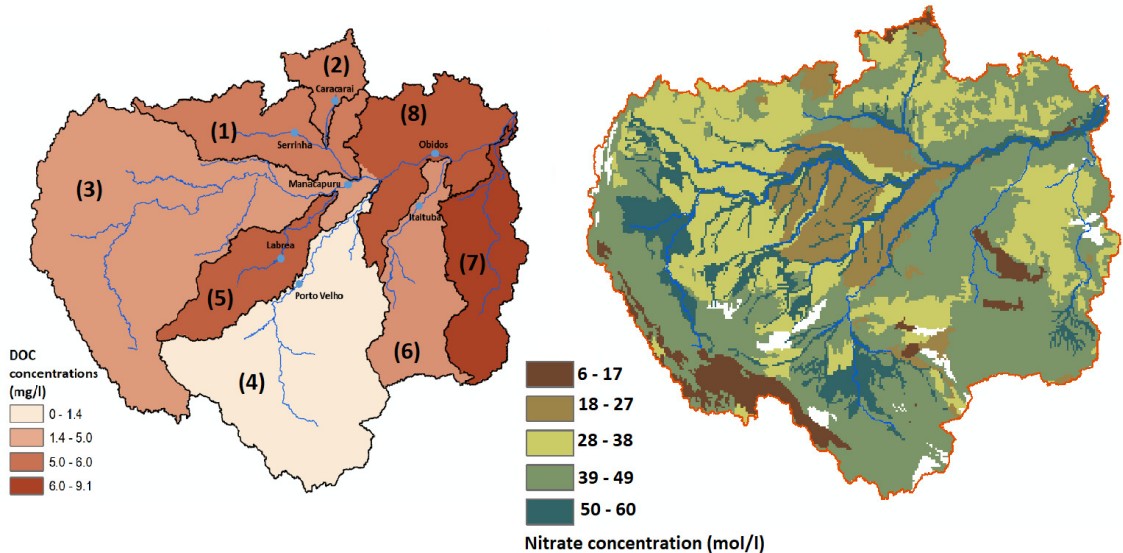

**Figure 3.** Map of the spatial inputs of the denitrification model. DOC contents in mg/L mapped over each sub-basin of the main streams (January) with local observation gauging stations in blue circles (Left). The Amazon watershed is divided into 8 major sub-basins: (1) the Negro basin, (2) the Branco basin, (3) the Solimoes River and its tributaries, (4) the Madeira basin, (5) the Purus basin, (6) the Tapajos basin, (7) the Xingu basin and (8) the section between Manaus and the mouth of the Amazon River. $NO_3^-$ contents (mol/L) of the watershed over FAO's types of soils (Right).

are supposed not to modify significantly the amount of $NO_3^-$ contained in the soil solution. Indeed, the concentration of $NO_3^-$ in the river is negligible to the concentration of riverine aquifers (Sánchez-Pérez et al., 2003). We consider that the DOC in the soil is directly brought by streams so the amount of DOC included in soils is set up to the values of the streams. Most of the organic carbon is transported from alluvial sediments or brought by streams during flooding events (Peter et al., 2012).

5 Because of the supersaturation of $p_{CO_2}$ in groundwater (Davidson et al., 2010), we consider that the gases produced during denitrification are entirely emitted to the atmosphere . Overall, denitrification was calculated as:

$$D_{NO_3} = R_{NO_3} \cdot SWAF \cdot Q_{wa} \tag{5}$$

where $D_{NO_3}$ is the net denitrification in mol month$^{-1}$, $R_{NO_3}$ is the denitrification rate in mol month$^{-1}$ L$^{-1}$, SWAF is the fraction of land covered with open waters and $Q_{wa}$ is the water storage capacity for each type of soil (L) retrieved from the

10 FAO soil database. In summary, the model requires the inputs and parameters for: (1) the $NO_3^-$ concentration for each type of soil (mol/L), (2) the DOC concentrations of the streams that overflow, extended to the associated sub-basin and (3) the extent

of inundated surfaces. The model simulations were applied over the Equal-Area Scalable Earth Grids version 2 (EASEv2) nodes at daily scale from January $1^{st}$ 2011 to December $31^{th}$ 2015 and monthly maps were then generated. Note that in order to assess the denitrification only occurring in wetlands, the minimum SWAF value recorded during the period (2011-2015) is subtracted to each month simulation, as it accounts as a residual artefact of streams.

## 3 Results

### 3.1 Spatial and temporal patterns of denitrification over the Amazon basin

Denitrification and emissions of $CO_2$, $N_2O$ are simulated for each month from 2011 to 2015. Figure 4 shows the yearly average maps of denitrification, $CO_2$ and $N_2O$ emissions over the Amazon basin. The three major hot spots which correspond to the major floodplains of the Amazon Basin are identified.

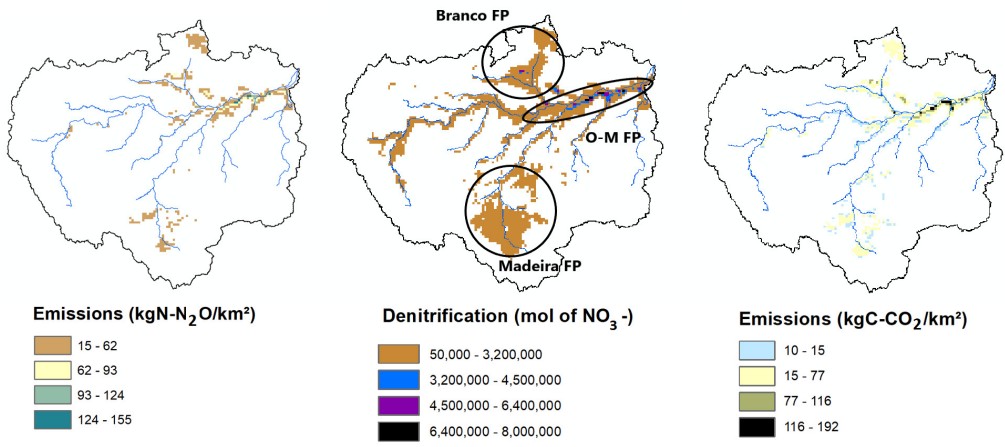

**Figure 4.** Spatial representation of $N_2O$ emissions (kgN-$N_2O$/km²), denitrification (mol of $NO_3^-$) and $CO_2$ emissions (kgC-$CO_2$/km²) summed over the year 2013. The locations of the main floodplains (hot spots) are outlined in the denitrification map.

Denitrification time series over the entire Amazon basin (Fig. 5) show that the denitrification process leads to similar temporal patterns of $CO_2$ and $N_2O$ emissions at the basin scale. From November to March the denitrification and the emissions become active with the increase of $NO_3^-$ denitrified in the basin. During the first months, until December, the activation is slow and mild. It then increases in the following months and peaks in March at $1.16 \times 10^9$ kg of N-$NO_3^-$ denitrified, $2.15 \times 10^8$ kg of C-$CO_2$, $1.00 \times 10^8$ kg of N-$N_2O$. Between March and June, the denitrification and the emissions are steady and fluctuate respectively around $9.51 \times 10^8$ kg of N-$NO_3^-$ denitrified, $2.04 \times 10^8$ kg of C-$CO_2$, $9.51 \times 10^7$ kg of N-$N_2O$. Finally, it is observed from June to October that the processes inactivate at a slower rate (-33%) than activation. Subsequently, the decreasing trend shifts and tops in August. Values registered in September are lower than in August, and yet in the years 2011, 2012 and

2015, these were similar. The decreasing trend reaches eventually a minimum peak in November at $1.96 \times 10^8$ kg of N-NO$_3^-$ denitrified, $4.20 \times 10^7$ kg of C-CO$_2$, $1.96 \times 10^7$ kg of N-N$_2$O.

The same pattern of denitrification repeats every year during the period of the study (2011-2015). We find that the denitrification process can be separated into three phases. First, an activation phase that is triggered by the increase of the flooded areas and the increase in the microbiological activities. Second, a stabilization phase which corresponds to a maximum denitrification rate and a peak in microbiological activities. And third, a deactivation phase which corresponds to the retreat of the inundation which also reduced the microbiological processes of denitrification. Note that this conclusion is not independent of the selected model implementation and associated assumptions. Additionally, it shows more precisely three hot moments in March, June and August of each year. The first two hot moments, in March and June, are maximum area peaks. During these months, despite observing a low activity over the watershed (below $8.70 \times 10^5$ kg of N-NO$_3^-$ denitrified per pixel), the extent of surfaces undergoing denitrification is the highest. On the contrary, the August hot moment is mainly due to a particularly strong denitrification between Obidos and Manaus with peaks of 6.16 and $7.20 \times 10^6$ kg of N-NO$_3^-$ denitrified. CO$_2$ emissions average $1.75 \times 10^8$ kg of C-CO$_2$ per month over the basin. N$_2$O emissions fluctuate around $6.52 \times 10^7$ kg of N-N$_2$O per month from the watershed.

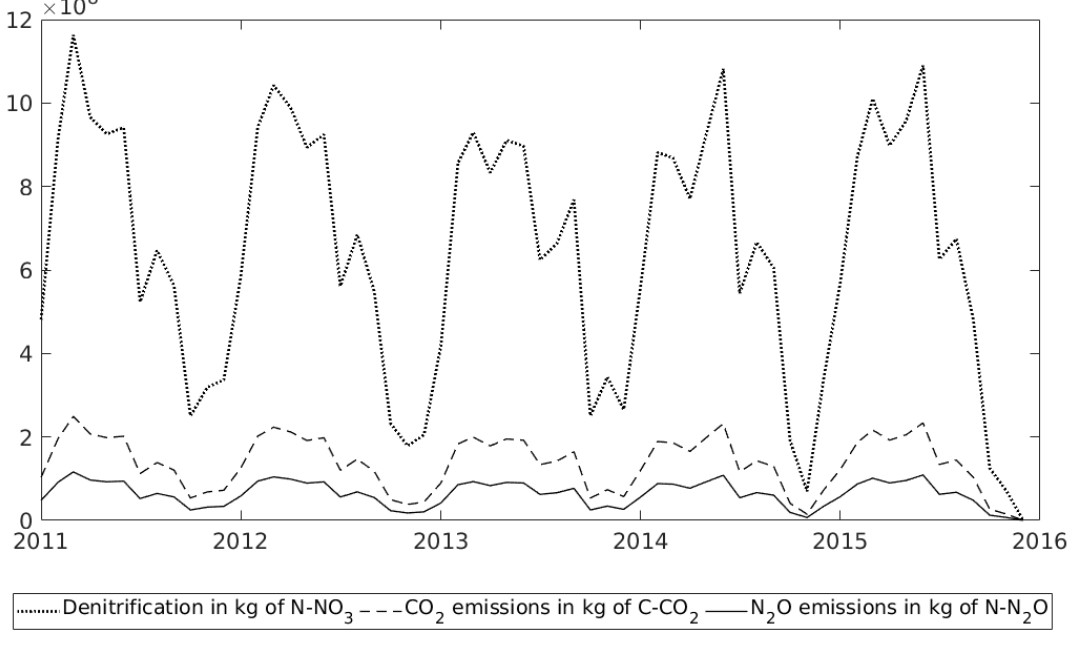

**Figure 5.** Monthly denitrification (kgN-NO$_3$), CO$_2$ (kgC-CO$_2$)and N$_2$O (kgN-N$_2$O) emissions over the entire Amazon watershed for the period 2011 - 2015.

## 3.2 Denitrification, $CO_2$ and $N_2O$ emissions: focus on the three main Amazon floodplains

The temporal patterns of the processes over the entire basin and throughout the whole period are unique in each floodplain. In fact, the three floodplains do not become active/ inactive at the same time and do not reach their maximum potential activity at the same moment either. Figure 6 shows the monthly behaviour of $N_2O$ emissions over the basin and for each floodplain together. The denitrification as well as the $CO_2$ and $N_2O$ emissions follow the same patterns but on different proportions. The results of the model provide the following inferences:

- The O-M FP follows the same pattern as the watershed trend and is mainly active between March and June but it never becomes totally inactive during the October – December period. It undergoes an average denitrification of $2.20 \times 10^8$ kg of $N\text{-}NO_3^-$ and emissions of $4.78 \times 10^7$ kg of $C\text{-}CO_2$ and $2.23 \times 10^7$ kg of $N\text{-}N_2O$.

- The Madeira FP follows the same pattern as the O-M FP. However, it becomes active in October and reaches on average its maximum emissions in March with $2.93 \times 10^8$ kg of $N\text{-}NO_3^-$ denitrified, $6.28 \times 10^7$ kg of $C\text{-}CO_2$, $2.93 \times 10^7$ kg of $N\text{-}N_2O$. The intensity of the processes decreases rapidly after. A maximum peak is usually observed afterwards in June with $3.03 \times 10^8$ kg of $NO_3^-$ denitrified, $6.49 \times 10^7$ kg of $C\text{-}CO_2$ and $3.03 \times 10^7$ kg of $N\text{-}N_2O$. The Madeira FP denitrification is almost inactive between July and October with emissions below $5.17 \times 10^7$ kg of $N\text{-}NO_3^-$ denitrified, $1.11 \times 10^7$ kg of $C\text{-}CO_2$ and $5.17 \times 10^6$ kg of $N\text{-}N_2O$.

- The Branco FP emissions are the least constant of the three floodplains even though a general pattern can be observed. The floodplain becomes active in January but the activation is slow and the denitrification is low until April (less than $1.70 \times 10^8$ kg of $N\text{-}NO_3^-$) as well as the emissions ($4.00 \times 10^7$ kg of $C\text{-}CO_2$ and $1.70 \times 10^7$ kg of $N\text{-}N_2O$). Afterwards, the processes intensity increases and tops in May (2011, 2012, 2013) / June (2014 and 2015) and September 2013 at $4.06 \times 10^8$ kg of $N\text{-}NO_3^-$, $8.71 \times 10^7$ kg of $C\text{-}CO_2$, $4.06 \times 10^7$ kg of $N\text{-}N_2O$. The floodplain is the least active from October to February/March with denitrification and emissions barely reaching $1.20 \times 10^8$ kg of $N\text{-}NO_3^-$ and $2.50 \times 10^7$ kg of $C\text{-}CO_2$, $1.20 \times 10^7$ kg of $N\text{-}N_2O$ respectively.

The detailed functioning of each floodplain explains the general pattern observed for the processes. The O-M FP drives the general trends of the total denitrification, $CO_2$ and $N_2O$ emissions of the watershed and the three different phases: activation, stabilization and deactivation. The March peak is mainly due to the Madeira FP reaching a maximum of activity. The June peak is also attributed to the Madeira floodplain for the years 2011, 2012 and 2013. The peak in 2014 is due to the combined contributions of the Branco FP and the Madeira FP topping activities, whereas in 2015 only the Branco FP is contributing. The August peak is again due to the rising of the O-M FP and the Branco FP activity.

Figure 7 shows the monthly contribution of each floodplain to the total denitrification as well as the average monthly denitrification over the basin for the period 2011-2015. Overall, the three floodplains contribute to 80% of the basin denitrification. From January to March it is mainly supported by the O-M FP and the Madeira FP, whereas from July to November it is due to the O-M FP and the Branco FP activity. In April, May, June and December the involvement of the floodplains is similar. We ran an ANalysis Of VAriance (ANOVA) and a post-hoc analysis to determine the contribution of each floodplain to the basin

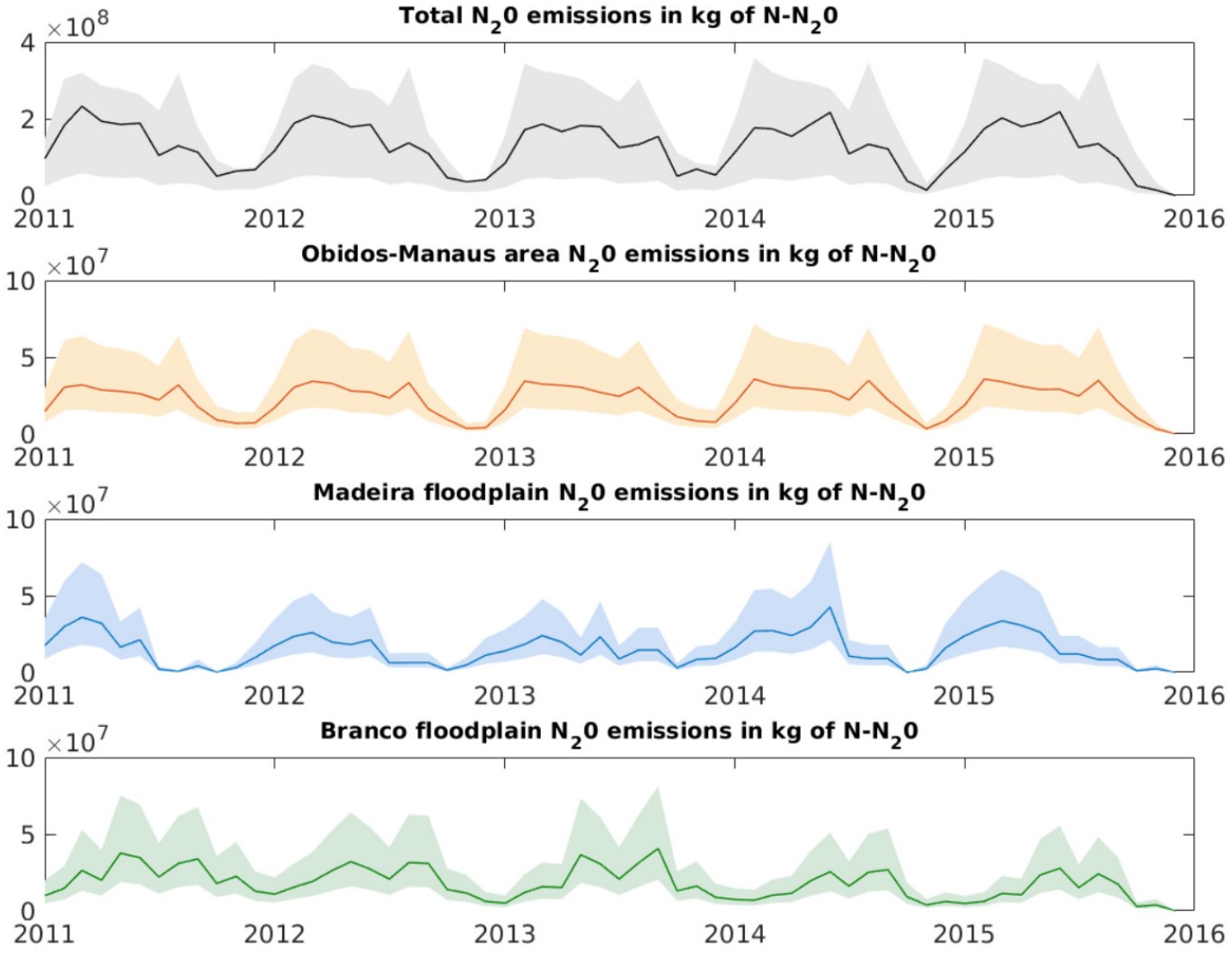

**Figure 6.** Monthly time series of $N_2O$ emissions over the basin (black), for the O-M FP (yellow), for the Madeira FP (blue) and for the Branco FP (green) over the period (2011-2015). The lines represent the emissions for a $N_2O / N_2$ of 0.1 whereas the coloured areas refer to the potential range of the ratio (0.05 - 0.2). Denitrification and $CO_2$ emissions follow the same patterns but with a scale factor of times 10 for denitrification and times 2 for $CO_2$.

denitrification. The results showed two different groups (p.value = $1.35 \times 10^{-8}$, alpha = 5%). The first group is constituted by the O-M FP which is the main source of denitrification for the basin and provides 38% of the processes on average. The second group is constituted by the Branco FP and the Madeira FP. They contribute similarly to the processes (on average 25% and

21% respectively) The same conclusions can be made for the $CO_2$ and $N_2O$ emissions.

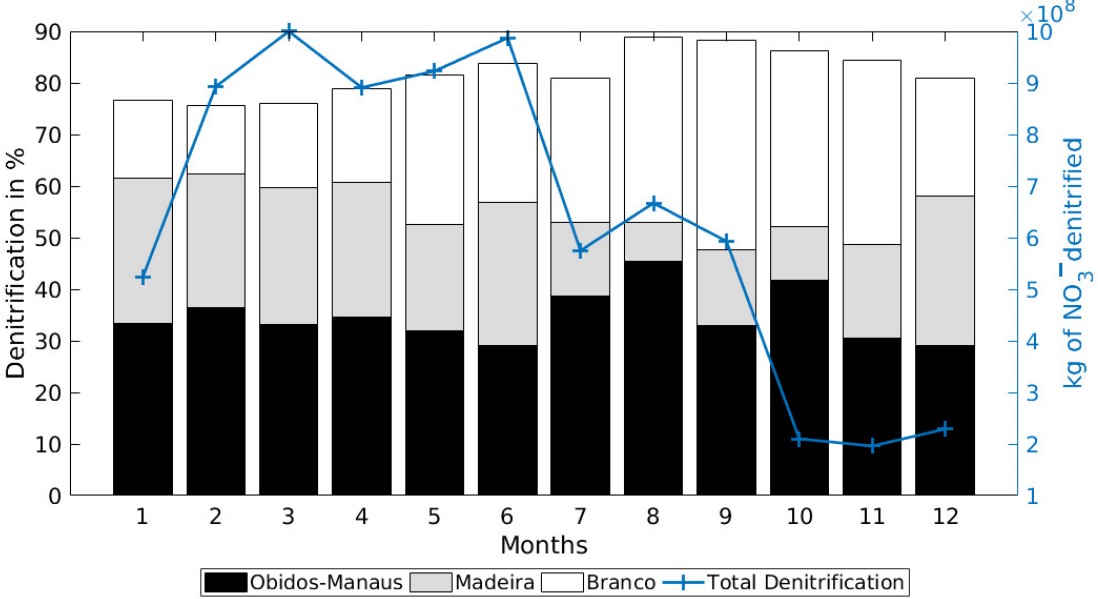

**Figure 7.** Average monthly contribution of each floodplain: the O-M FP (black), the Madeira FP (grey), Branco FP (white) to the Amazon total denitrification. The residual contribution from the 100% is associated with the other wetlands in the basin. The blue line represents the average monthly denitrification for the period of the study and it shows the main trend observed over the Amazonian watershed.

### 3.3 Greenhouse gases emissions from the Amazonian wetlands

Table 1 depicts the yearly emissions of $CO_2$ and $N_2O$ over the Amazon basin and the three main floodplains. Emissions of $CO_2$ from denitrification are twice as much higher than $N_2O$ emissions over the basin. The yearly emissions of $CO_2$ from 2011 to 2015 over the Amazon basin show significant low interannual differences (Kruskal-Wallis p.value = 0.9929). The same conclusion is drawn for the yearly $N_2O$ emissions. On average, flooded areas emit $2.20 \times 10^9$ kg C-$CO_2$ per year and $1.03 \times 10^9$ kg N-$N_2O$ per year by denitrification from the natural $NO_3^-$ pool of the watershed.

During that period, the O-M FP is the floodplain which contributes the most to the emissions for the two gases. The dynamics of the Madeira FP and the Branco FP changed in 2014. Indeed from 2011 to 2013, the Branco FP roughly emitted twice as many gases than the Madeira FP. This trend shifted in 2014 with the involvement of the Madeira FP becoming more important in term of emissions than the Branco FP. At a yearly basis, the whole Amazon basin undergoes a denitrification of about $1.03 \times 10^{10}$ kgN/ha/yr.

**Table 1.** Average yearly $CO_2$ emissions in kgC-$CO_2$, $N_2O$ emissions in kgN-$N_2O$ and $N_2$ emissions in kgN for the Amazon basin and the three main floodplains. The value are calculated for a $N_2O$ / $N_2$ ratio of 0.1.

| Wetland | Area (ha) | $CO_2$ (kgC) | $N_2O$ (kgN) | $N_2$ (kgN) |
|---|---|---|---|---|
| Amazon basin | $5.7 \times 10^8$ | $2.20 \times 10^9 \pm 2.75 \times 10^8$ | $1.03 \times 10^9 \pm 2.57 \times 10^7$ | $9.26 \times 10^9 \pm 2.57 \times 10^8$ |
| Obidos - Manaus FP | $2.5 \times 10^7$ | $7.63 \times 10^8 \pm 9.94 \times 10^7$ | $3.56 \times 10^8 \pm 9.28 \times 10^6$ | $3.21 \times 10^9 \pm 9.28 \times 10^7$ |
| Madeira FP | $3.7 \times 10^7$ | $4.79 \times 10^8 \pm 2.65 \times 10^8$ | $2.24 \times 10^8 \pm 2.47 \times 10^7$ | $2.01 \times 10^9 \pm 2.47 \times 10^8$ |
| Branco FP | $6.78 \times 10^7$ | $5.57 \times 10^8 \pm 6.17 \times 10^8$ | $2.6 \times 10^8 \pm 5.75 \times 10^7$ | $2.34 \times 10^9 \pm 5.75 \times 10^8$ |

## 3.4 Denitrification and trace gas emissions anomalies

During the period of the study, major meteorological events were recorded over the Amazon basin. On the one hand, the year 2011 was a year influenced by La Niña (Moura et al., 2019). La Niña periods lead to wetter weather conditions in South America. From October 2013 to March 2014, heavy rainfalls were documented on the Madeira region and caused extreme flooding in this region and nearby Obidos. On the other hand, September 2015 marked the begging of an "El Niño" episode. In South America and the Amazon, El Niño produces drier weather conditions.

Fig.8 shows the monthly anomalies of denitrification observed over the Amazon watershed from 2011 to 2015. Anomalies were determined by first calculating the mean value for each month across the period 2011-2015. This mean value was then subtracted from each corresponding month in the series. Positive anomalies show an intense denitrification whereas negative anomalies show a denitrification lower than the average. Examining the anomalies of the watershed and the floodplains show that during La Niña year and the heavy precipitations period, most of the anomalies are positive especially for the first months (66% - 66% for the basin denitrification, 16% - 83% for the O-M FP, 25% - 33% for the Madeira FP and 100% - 50% for the Branco FP respectively). During El Niño episode, all the anomalies are negative. Nevertheless, el Niño is the only meteorological event that has a significant effect on the processes (p.value= $4.40 \times 10^{-3}$). Moreover it impacts the three floodplains (p.value= $3.43 \times 10^{-4}$). Months undergoing the El Niño episode show a reduction of 27.7% from the average values.

Extreme events do not have a consistent impact on the whole basin. Table 2 sums up the spatial denitrification for the Amazon basin and the three floodplains at a yearly scale. Extreme meteorological events do not impact the denitrification and trace gases emissions at the basin scale. The average yearly denitrification rates for the whole basin, the O-M FP and the Madeira FP show no clear trend between 2011 and 2015. For the Branco FP, a decreasing trend was identified during the study period. From 2011 to 2015 the simulated average yearly denitrification for the Branco FP drops by a factor two.

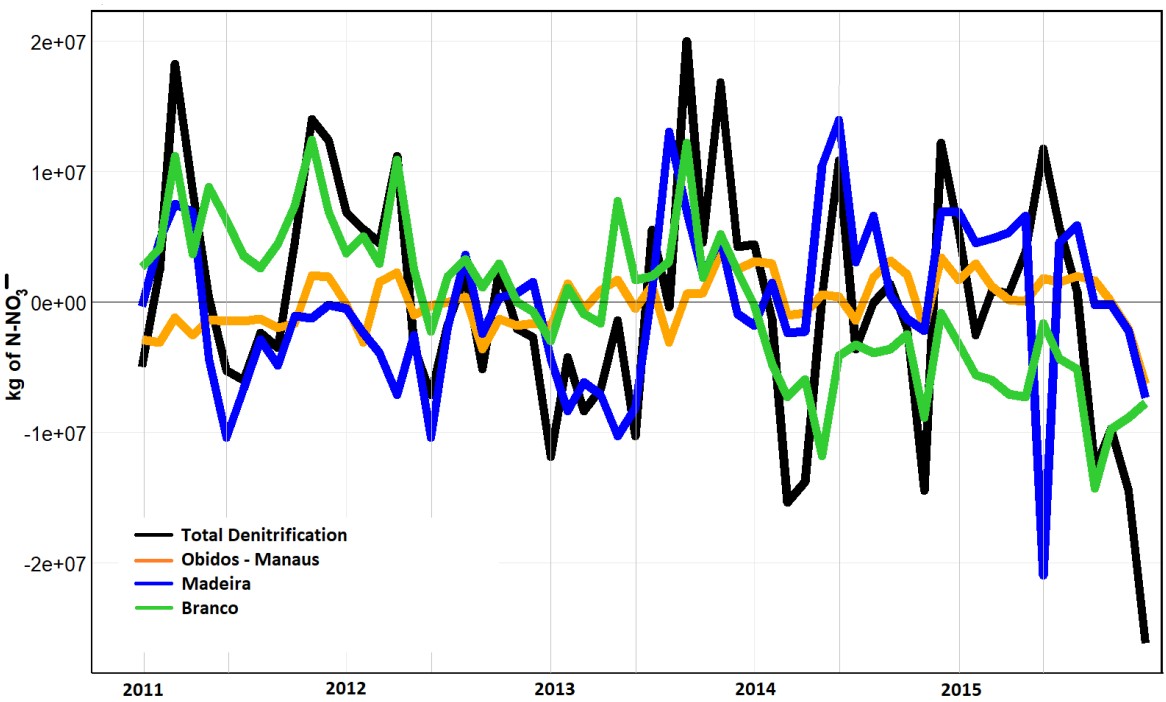

**Figure 8.** Monthly anomalies at the basin and main floodplains scale for denitrification throughout the period (2011-2015).

**Table 2.** Yearly denitrification in kgN/ha/yr for the whole basin and the three major floodplains from year 2011 to 2015.

| Denitrification (kgN/ha/yr) | 2011 | 2012 | 2013 | 2014 | 2015 |
|---|---|---|---|---|---|
| Basin | 18.4 | 18.0 | 17.9 | 17.5 | 17.2 |
| O-M FP | 137.3 | 140.6 | 144.9 | 146.9 | 142.7 |
| Madeira FP | 57.4 | 56.3 | 53.3 | 67.4 | 67.7 |
| Branco FP | 48.5 | 43.0 | 43.0 | 31.4 | 28.3 |

## 4 Discussion

### 4.1 Determining key factors of the denitrification

A sensitivity analysis of the parameters of the denitrification Eq. (1) was performed. $k_{POC}$ can range from $0.15 \times 10^{-6}$ to $1.10 \times 10^{-4}$ which leads to a yearly denitrification 46% lower and 18% higher than the initial values respectively. $k_{DOC}$ range from $1.00 \times 10^{-4}$ to $1.22$ which leads to values of denitrification 94% lower and $130\,000\%$ higher respectively. It follows that for the Amazon Basin $k_{DOC}$ is evaluated as more sensitive than $k_{POC}$. Also, the $NO_3^-$ related part of the denitrification equation was analysed. $NO_3^-$ are relatively abundant in the watershed's soils and it is noticeable that $k_{NO_3}$ is negligible compared to $NO_3^-$ though $\lim_{NO_3^- \to \infty} \frac{[NO_3^-]}{[k_{NO_3} + NO_3^-]} = 1$. $NO_3^-$ is a non-limiting factor of denitrification for the Amazon basin. Overall, the

denitrification equation currently depends on four variables: POC, DOC, NO$_3^-$ and SWAF. Overall, the main driving variables of the denitrification model are SWAF and DOC.

**Table 3.** Overall denitrification in kgN/ha/yr, mean and standard deviation of the SWAF and DOC (mg/L) values for the three floodplains

| Floodplain | Denitrification | DOC | | SWAF | |
|---|---|---|---|---|---|
| | | Mean | Standard deviation | Mean | Standard deviation |
| O-M FP | 142.5 kgN/ha/yr | 5.65 mg/L | 2.45 mg/L | 3.3% | 0.12% |
| Branco FP | 38.8 kgN/ha/yr | 8.93 mg/L | 2.87 mg/L | 1.4% | 0.27% |
| Madeira FP | 60.4 kgN/ha/yr | 2.26 mg/L | 2.45 mg/L | 1.7% | 0.17% |

Table 3 depicts for the O-M FP, the Madeira FP and the Branco FP the effective denitrification over the 2011-2015 period in kgN/ha/yr as well as the average and standard deviation values of DOC concentration in mg/L and SWAF index. The denitrification values show that all three floodplains are particularly active systems in term of processing organic matter and NO$_3^-$. The O-M FP is an active floodplain in term of denitrification potential with an average annual intensity of 142.5 kgN/ha/yr. The DOC show that the Branco FP is the highest floodplain in terms of DOC concentration with an average of 8.93 ± 2.87 mg/L, followed by the O-M FP with 5.65 ± 2.45 mg/L and the Madeira FP 2.26 ± 2.45 mg/L. Similar to the DOC, the average and standard deviation of the SWAF values were extracted from the daily observations over the 2011-2015 period. The ranked order of the floodplains for the SWAF component is similar to the denitrification one. This result strengthens the importance of Earth Observation (EO) based monitoring of water bodies for determining inundated surfaces patterns and intensities and their impact on biochemical processes. Eventually, the differences of denitrification intensity observed for the three floodplains are the combined effect of the variations of the DOC concentrations and the SWAF. As a matter of fact, DOC determines the average maximum denitrification rate of a floodplain, whereas the SWAF value is the main driving factor of the model which reveals the actual denitrification. Overall, the denitrification rate (Eq. 1) should be considered as a combination of a potential rate function (provided by DOC and POC) and limitation functions provided by the peculiar environmental conditions.

## 4.2 Comparing to physically-based models

The N$_2$O emissions at large scale were compared to results of the N$_2$O Model Inter-comparison Project (NMIP) project (Tian et al., 2018) model, more particularly the Dynamic Land Ecosystem Model (DLEM) (Xu et al., 2017), the Vegetation Integrative SImulator for Trace gases (VISIT) (Ito and Inatomi, 2012) and the Organising Carbon and Hydrology In Dynamic Ecosystems - Carbon Nitrogen (ORCHIDEE-CN) (Zaehle and Friend, 2010) models. These models consider the N$_2$O emissions from nitrification and denitrification, where in our case only denitrification during flooding is considered. In our case, $k_{POC}$ and $k_{DOC}$ are the mineralization rate parameters. They describe the kinetic processing of organic matter into POC and DOC respectively. The organic matter processing is performed by microbial communities. Therefore, environmental conditions such as temperature and soil pH have a direct influence on bacterial activity and turnover. The cumulated impact of temperature, soil pH and microorganisms activity is accounted for indirectly in our approach through the parameters $k_{POC}$ and $k_{DOC}$

described in Eq. 1 (Peyrard et al., 2010; Sun et al., 2017).

During the period 2011-2015 those models evaluated emissions of $N_2O$ from the Amazon basin at about 0.14 gN/m²/yr. Our model simulates emissions of $N_2O$ at roughly $0.18 \pm 4.4 \times 10^{-3}$ gN/m²/yr over the basin. The peculiar emission of the $1.3 \times 10^{11}$ m² wetlands system represents $0.81 \pm 0.02$ gN/m²/yr. We can observe that our model gets a total higher estimation of the emissions of $N_2O$ at a rate of 28% than the other models with 80% of them (0.14 gN/m²/yr) originates from the three main floodplains; the O-M FP, the Madeira FP and the Branco FP. In term of input data, our model as well as DLEM, VISIT and O-CN use climate data, soil types and inundated fractions/surfaces. A divergent point is how nitrogen pool is calculated. We consider it as being produced by the organic matter mineralization and a maximum nitrification, whereas the other models compute it from nitrogen deposition. Moreover, they also take natural vegetation, swamps delineation (O-CN) and land cover as input data while we only focus on wetland types. These models assess $N_2O$ emissions based on the processes of the nitrogen cycle such as denitrification. Our model apprehends denitrification as a function of carbon and nitrate contents (DOC, POC and $NO_3^-$) and inundated surfaces (SWAF). As a result, these models do not fully distinguish the alluvial floodplain from other lands (Xu et al., 2017) and underestimate its effects (Ito and Inatomi, 2012). Thus our results bring us to conclude that current physically-based $N_2O$ emissions models are likely to slightly underestimate the contribution of wetlands in the global budget.

## 4.3 Wetlands and integrated ecosystem emissions

In this section, our model outputs for wetlands emissions are compared to local in situ measurements of the $N_2O$ and $CO_2$ ecosystem emissions. Table 4 summarizes the different results from in situ measurements for $N_2O$ and $CO_2$ and the closest simulation node from our simulation. We extracted the average simulated value of the period from the simulation node. When comparing the $N_2O$ with in situ campaigns performed by Koschorreck (2005), Keller et al. (2005) and Liengaard et al. (2014) at the different locations, the wetlands emissions from our study are roughly ower from a factor $10^2$ of the integrated ecosystem observed emissions. This difference comes from different spatial and temporal scales for both the in situ measurements and our model. To decrease the variability, we extracted the maximal pixel value simulated during the period of the study. On average, in situ measurements return emissions of about $4.9 \times 10^7$ gN/km²/yr while our highest simulation value estimated an emission of about $2.6 \pm 1.3 \times 10^7$ gN/km²/yr.

$CO_2$ emissions at local in situ measurements (Keller et al., 2005) as well as to broader measurements (Richey et al., 2002) are compared to our model's outputs. Our wetlands estimations are considerably lower ($10^4$) than the integrated ecosystem observations. As expected, even though $CO_2$ emissions from wetland denitrification are about $2.16 \times 10^9$ kgC-$CO_2$ per year over the Amazon basin, these emissions are negligible when compared to the full ecosystem carbon emissions (Cole et al., 2007; Davidson et al., 2010). Overall, $CO_2$ emissions from denitrification over the whole Amazon basin contribute with 0.01% of the carbon emissions of the watershed. Most of the $CO_2$ emissions over the Amazon are attributed to processes such as organic matter respiration from biomass and little contributions from wetlands. Vicari et al. (2011) showed that the change of wetlands into forested area can increase the carbon emissions drastically. In this context and in the light of the results obtained

in this paper one can conclude that in case of very dry natural events or intense anthropogenic changes of the land-cover the carbon budget of the once wetland areas and now non-inundated surfaces will greatly increase.

**Table 4.** Comparison of the values estimated by our study and the literature for emissions of $CO_2$ (gC/km²/yr) and $N_2O$ (gN/km²/yr).

| Paper | Gas measured | Site | Ecosystem in situ obs. | Modelled wetlands |
|---|---|---|---|---|
| Koschorreck (2005) | $N_2O$ | Manaus plateau | $5 \pm 7.5 \times 10^6$ | $2.4 \pm 1.1 \times 10^4$ |
| Keller et al. (2005) | $N_2O$ | Santarem | $8.6 \pm 0.7 \times 10^6$ | $5.2 \pm 0.9 \times 10^4$ |
| Liengaard et al. (2014) | $N_2O$ | Rio Solimoes | $4.4 \times 10^7$ | $5.7 \pm 2.8 \times 10^5$ |
| Liengaard et al. (2014) | $N_2O$ | Rio Cupea | $8.3 \times 10^7$ | $5.7 \pm 2.8 \times 10^5$ |
| Liengaard et al. (2014) | $N_2O$ | Rio Amazonas | - | $9.3 \pm 4.6 \times 10^5$ |
| Liengaard et al. (2014) | $N_2O$ | Iagarapé de Paracuba | $1.9 \times 10^7$ | $1.1 \pm 0.6 \times 10^5$ |
| Liengaard et al. (2014) | $N_2O$ | Rio Tapajos | $1.9 \times 10^7$ | $1.5 \pm 0.7 \times 10^6$ |
| Liengaard et al. (2014) | $N_2O$ | Rio Mucajai | $7.8 \times 10^7$ | $2.1 \pm 1.1 \times 10^5$ |
| Richey et al. (2002) | $CO_2$ | Amazon River wetlands | $6 \pm 0.3 \times 10^7$ | $4.4 \pm 2.5 \times 10^3$ |
| Keller et al. (2005) | $CO_2$ | Santarem | $5.7 \pm 0.6 \times 10^7$ | $1.6 \pm 0.9 \times 10^3$ |

## 4.4 The Amazonian wetlands emissions versus Tropical and temperate wetlands

We put in perspective the Amazonian wetlands emissions to a variety of wetland ecosystems such as the Congo basin, rice
paddies of south-eastern Asia, the Garonne (France) and the Rhine (Europe) rivers with each possessing peculiar features. The Congo basin can be considered, like the Amazon, as a pristine ecosystem regarding agricultural nitrogen inputs. On the contrary, rice paddies regions are territories with intensive agricultural activities, high $NO_3^-$ fertilization and undergo several flood events per year. Both the Congo basin and the rice paddies regions are part of the tropical region, like the Amazon basin. The $N_2O$ emissions from the Amazon and the Congo basins are comparable. Our results for the Amazon and the ones exposed
in Tian et al. (2018) for the Congo show emissions of 0.18 gN/m²/yr. The two watersheds are pristine for agricultural nitrogen inputs and located toward the same latitudes, so relatively similar emissions of $N_2O$ are expected. On the contrary, rice paddies shoot up with emissions of about 0.28 gN/m²/yr. This is explained by the impacts of agricultural inputs and successive flooding on wetland ecosystems that increase the amount of greenhouse gases emitted. The Garonne and the Rhine rivers catchments are in temperate regions under high agricultural pressures. The Garonne river, one of the main fluvial systems in France, is
525 km long draining a 55 000 km² area into the Atlantic Ocean. The large range of altitudes and slopes within the watershed leads to a diversity of hydrological behaviours. The typical alluvial plain starts from its middle section and is about 4 km wide. The riparian forest and poplar plantations cover the first 50-200 m from the riverbank, beyond which lies agricultural land that accounts for 75% of the total area. The Rhine river, one of the main fluvial systems in Germany, is 1,233 km long draining a 198 000 km² area from Switzerland to the North Sea. The average denitrification reaches $132.52 \pm 3.9$ kgN/ha/yr Sun et al.
(2017) and 653 kgN/ha/yr Sánchez-Perez et al. (1999) for the Garonne's and Rhine's floodplains respectively. The average rate of denitrification for the Amazon basin is $17.8 \pm 0.4$ kgN/ha/yr which is far less than values observed in European catchments.

As a comparison, the Òbidos - Manaus floodplain (table 2) denitrification potential is equivalent to the Garonne river. Overall, the Amazon wetland ecosystem can be regarded as a not-very active greenhouse gases emitting system compared to other ecosystems of the tropical region. Moreover, our results show that the O-M FP possesses the same denitrification potential as a $NO_3^-$ polluted temperate ecosystem.

## 4.5   Limitations of the current approach

The findings of this study have to be seen in the light of some limitations. First, the sampling resolution of the input data can induce bias. The SWAF product tends to underestimate water surface extents variability and land cover identification due to the coarse resolution of 25 km x 25 km. Second, the use of uniform $k_{POC}$ and $k_{DOC}$ values limits the capabilities of the model to fully consider the impact of the spatial variability of both geophysical and biological variables. Third, an average $N_2O$ / $N_2$ ratio of 0.1 was set up for the study. It varies depending on several conditions as soil properties, land cover, temperature and more. Thus a precise and spatial estimation of the ratio was not relevant due to the low resolution of our input data and the lack of in field measurements. Fourth, as highlighted by the present study, the lack of in situ measurements of $N_2O$ emissions over tropical wetlands specifically increases the uncertainties and equifinalities for the calibration of model parameters and validation. Fifth, considering the dynamics of the activation-stabilization-deactivation of the denitrification, they can be more precisely assessed if variables like water surface temperatures and water depth were added in the future. These variables can inform on the speed at which the activation and deactivation of the microbiological process of denitrification are triggered. Future studies should concentrate on adding more remotely sensed geophysical variables at the adapted spatial resolution (Parrens et al., 2019), taking into account the fact that flooding actually sustains the different processes. Sixth, denitrification and dissimilatory nitrate reduction to ammonium (DNRA) are two natural processes for $NO_3^-$ reduction. In their, review Rütting et al. (2011) state that DNRA competition for $NO_3^-$ should be considered for some ecosystems which did not include aquatic ecosystems. They added that more studies are needed for terrestrial aquatic ecosystems based on Burgin and Hamilton (2007). Tiedje et al. (1982) showed that under $NO_3^-$ limiting and strongly reducing conditions, DNRA has the advantage over denitrification. Sotta et al. (2008) estimated at 12-50% the reduction of $NO_3^-$ from DNRA in low land Brazilian forest but in non-flooded periods. In our case, $NO_3^-$ is non-limiting, thus we do not need to take into account the impact of $NO_3^-$ loss from DNRA. Moreover, since estimates of the DNRA direct contribution to $N_2O$ emissions is about 1% (Cole, 1988) and considering the uncertainty and errors linked to the modelling of denitrification in the wetlands of the whole Amazon basin the DNRA processes were not considered. Finally, in our study we focused on denitrification solemnly. In order to provide a complete nitrogen budget for the whole Amazon basin, future studies will need to complexify the proposed methodology by integrating additional biogeochemical processes (DNRA, nitrification,. . . ) and physically relative datasets (soil temperature, soil moisture,. . . ) in order to extend the approach to non-flooded periods and other ecosystems.

# 5 Conclusions

The main objective of the study is to quantify and assess $CO_2$ and $N_2O$ emissions over the Amazonian wetlands during flooding periods. To achieve these goals we design a data-based methodology that relies on modelling and remote-sensing products. It aims to estimate emissions linked to denitrification at large scale. The model parametrisation was justified by results from several published papers. It appears that denitrification mainly relies on DOC contents in the watershed. The study also contributes to better understand the functioning of the major floodplains of the Amazon Basin and their respective involvement in the Amazon carbon and nitrogen budget. It transpires that the most active floodplain is the Òbidos-Manaus, which is responsible for the majority of the processes. Each floodplain possesses its own functioning that depends on rainfalls and the hydrology of the floodplain's river. Overall, the results appear quite alike to other large scale models; especially for $N_2O$ emissions. $CO_2$ emissions from denitrification account for 0.01% of the Amazon carbon budget and represent a fraction of $3.5 \times 10^{-6}$ of the global $CO_2$ emissions (natural and anthropogenic). When we compare our simulated $N_2O$ emissions from Amazonian wetlands to other estimations over the Amazon basin we find that our estimations are higher (+ 28%). For that reason, we emphasize the importance of distinguishing wetlands in nitrogen models as those areas are significant sources of $N_2O$ emissions. Key factors of the denitrification for the Amazon basin were identified in the study. From our model design perspective, we find that the denitrification for the Amazon wetlands is driven by first the extent of the flooded areas, which constrain the process and second by the DOC content in the soil solution, which determine the maximum denitrification potential. Future studies will concentrate on extending the current approach to other tropical basins, needless to say that local observations will be essential for the validation of such exercise and preferably over the same period of analysis. Data from future missions like SWOT will deliver water heights at 21 days of global coverage, which will improve the results of such studies through the integration of surfaces and volume information.

*Author contributions.* Ahmad Al Bitar, Sabine Sauvage, Marie Parrens, José-Miguel Sanchez-Pérez and Jérémy Guilhen conceived and designed the methodology and the algorithms. Jérémy Guilhen performed the analysis. Jean-Michel Martinez, Gwenael Abril and Patricia Moreira-Turcq provided the scientific expertise and corrections to the manuscript. Jérémy Guilhen and Ahmad Al Bitar wrotre the first draft. Marie Parrens did all the graphs. All authors wrote the final manuscript.

*Competing interests.* All co-authors declare that no competing interests are present

*Acknowledgements.* This work was funded by the Midi-Pyrénées' "Axe transversal cycle du Carbone, de l'Azote et gaz à effet de serre". The SWAF product was developed in the framework of the TOSCA SOLE and SWOT-downstream programs from CNES. We thank the HyBAm observatory network for providing the data needed for this study. We thank the FUI (Fonds Unique Interministériel) HYDROSIM Project (2018-2021). We thank the reviewers for their comments and their help in improving our manuscript.

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
