# Peer review of "Denitrification and associated nitrous oxide and carbon dioxide emissions from the Amazonian wetlands"

_Biogeosciences, 2020_

## Referee Comment (RC1) · Anonymous Referee #1 · 26 Feb 2020

The manuscript (ms) under review presents a new approach to estimate the emissions of $CO_2$ and $N_2O$ from the various floodplains along the Amazon River during 2011 and 2015. The approach combines satellite data (-> estimate of the water surface) and in-situ data with an empirical assessment of the nitrate reduction rate (i.e. denitrification) in the upper soil which in turn results in production and emissions of both carbon dioxide ($CO_2$) and nitrous oxide ($N_2O$). Although the presented results are of interest for a wider community, I have some concerns about the approach used for $NO_3^-$ reduction. Therefore, I can recommend publication only after major revisions.

Specific comments: - The $NO_3^-$ loss in the floodplains is solely attributed to denitrification. However, $NO_3^-$ loss in soils can also take place during dissimilatory nitrate reduction to ammonium (DNRA) (see e.g. Rutting et al., Biogeosci, 8, 2011). So, I am

wondering whether this could affect the estimate of $CO_2$ and $N_2O$ emissions. Please discuss. You may need to adjust the equations (1) to (4) to account for DNRA. Please replace denitrification with 'nitrate (or $NO_3$-) reduction' throughout the text.

- The amount of $N_2O$ produced is calculated with a constant $N_2O/N_2$ ratio of 0.1. You can do so but, unfortunately, there is no reference given for it (P6L22). Moreover, it should be discussed whether this ratio is constant or variable in the Amazonian wetlands. In other words, how representative is the selected value of 0.1? This is an important point because the choice of this ratio directly determines the magnitude of the $N_2O$ emissions and the variability of this ratio determines the 'error bar' of the $N_2O$ emission estimates.

- I am wondering why nitrification as a source of $N_2O$ under low $O_2$ is ignored. Please discuss.

- Title: Please note that the term 'carbon emissions' also includes emissions of methane and other C-containing gases which are not subject of the ms. Moreover, $NO_3$- could be lost during dissimilatory nitrate reduction to ammonium (DNRA), see my comment above. To this end, I suggest to modify the title to 'Nitrate reduction and associated carbon dioxide and nitrous oxide emissions from the Amazonian wetlands'.

- The central and lower panels of Figure 6 are meaningless. They show exactly the same graphs but scaled with a factor of 5 (for $CO_2$, see equation (4)) and 0.1 ( for $N_2O$; $N_2O/N_2=0.1$). Please remove.

- Please avoid using colloquial terms such as 'paramount' (see P2L11; P4L2; P18L9) or 'hot moments' (see Section 3.1). They should not be used in the context of a scientific text.

- Please have the text proof read by a native English speaker. There are many sentences and phrases which are odd.

- There are several (annoying) typos: mole should read mol (various places throughout

the text); 'og' should read 'of' in the caption of Fig. 5; N20 should read N2O (Fig. 5); 3rd column/2nd line in Tab. 1: there is something wrong with the exponent; 'anormalies' should read 'anomalies'(P13L15), etc.

- Please replace NO3 with NO3- (in the equations as well as throughout the text and figures)

---

## Author Comment (AC1) · 10 Apr 2020

On behalf of all the co-authors we thank the referee for the review and associated comments that helped us improve the manuscript.

The manuscript (ms) under review presents a new approach to estimate the emissions of $CO_2$ and $N_2O$ from the various floodplains along the Amazon River during 2011 and 2015. The approach combines satellite data (-> estimate of the water surface) and in situ data with an empirical assessment of the nitrate reduction rate (i.e. denitrification) in the upper soil which in turn results in production and emissions of both carbon dioxide ($CO_2$) and nitrous oxide ($N_2O$). Although the presented results are of interest for a wider community, I have some concerns about the approach used for $NO_3^-$ reduction.

Therefore, I can recommend publication only after major revisions.

Specific comments: "- The NO3- loss in the floodplains is solely attributed to denitrification. However, NO3- loss in soils can also take place during dissimilatory nitrate reduction to ammonium (DNRA) (see e.g. Rutting et al., Biogeosci, 8, 2011). So, I am wondering whether this could affect the estimate of CO2 and N2O emissions. Please discuss. You may need to adjust the equations (1) to (4) to account for DNRA. Please replace denitrification with 'nitrate (or NO3-) reduction' throughout the text."

In the current study, we consider the denitrification process during flood events. In these conditions NO3- is the limiting for both denitrification and DNRA. Moreover the DNRA contribution to N2O emissions is about 1% (Rüttin et al., 2011 from Cole (1988)) which is negligible considering the other sources of uncertainties at this scale. The environmental conditions for DNRA occurring (e.g NO3- limiting, high redox soil and high C/N) are not met in this case, thus the contribution of DNRA should be lower. Therefore, we are confident on our choice not to address the DNRA in this study. But this may be an issue that needs to be addressed for N2O budget at global scale in non-limiting conditions.

"- The amount of N2O produced is calculated with a constant N2O/N2 ratio of 0.1. You can do so but, unfortunately, there is no reference given for it (P6L22). Moreover, it should be discussed whether this ratio is constant or variable in the Amazonian wetlands. In other words, how representative is the selected value of 0.1? This is an important point because the choice of this ratio directly determines the magnitude of the N2O emissions and the variability of this ratio determines the 'error bar' of the N2O emission estimates."

We thank the reviewer for this essential comment that was discussed in the first stages of this work between co-authors. Indeed, a constant value of N2O/N2 can be argued and can be still accepted as mentioned by the referee. We actually based our estimates of this value from (Weier et al., 1992; Pérez et al., 2000).

We choose to keep a 0.1 ratio for N2O/N2 production. Our spatial resolution is coarse as we consider the flooded area over a 25 km x 25 km thus we don't take into account the landscape peculiarities. N2O/N2 ratio ranges from 0.05 to 0.2 and it is likely that several different ratios should be found within one pixel. Nevertheless, without any precise measurement on the actual ratio value and the different proportions we decided to set up an effective ratio of 0.1 (which is the most common for Amazonian wetlands : Pérez et al., 2000) for the whole watershed in order to not under/over estimate the emissions. In the manuscript, we only discuss about N2O values calculated from a 0.1 ratio (for better comprehension) but we added "error bars" corresponding to a 0.05 and 0.2 ration in the graphs. Comments in $ 2.4.1 P6 L20 and $ 4.5 P 17-18 were added to explain these choices.

"- I am wondering why nitrification as a source of N2O under low O2 is ignored. Please discuss."

Several studies showed that under anoxic conditions denitrification is the only source of N2O emissions (see Bollmann and Conrad 1998, Global Change Biology). The scoop of our paper is to specifically focus on denitrification and associated emissions, thus we did not take into account nitrification.

"- Title: Please note that the term 'carbon emissions' also includes emissions of methane and other C-containing gases which are not subject of the ms. Moreover, NO3- could be lost during dissimilatory nitrate reduction to ammonium (DNRA), see my comment above. To this end, I suggest to modify the title to 'Nitrate reduction and associated carbon dioxide and nitrous oxide emissions from the Amazonian wetlands'. "

We suggest to the editor a tittle change to"Denitrification with associated nitrous oxide and carbon dioxide emissions from the Amazonian wetlands"

"- The central and lower panels of Figure 6 are meaningless. They show exactly the same graphs but scaled with a factor of 5 (for CO2, see equation (4)) and 0.1 ( for N2O;

N2O/N2=0.1). Please remove. "

Fig.6 was changed to represent N2O emissions over the basin and the floodplains. Comments in the caption and the text P9 L6-8 were added to explain that denitrification, CO2 and N2O emission follow the same patterns with different values.

"- Please avoid using colloquial terms such as 'paramount' (see P2L11; P4L2; P18L9) or 'hot moments' (see Section 3.1). They should not be used in the context of a scientific text. "

We understand the worries of the reviewer on the potential use of colloquial terms though it was not the intention of the co-authors. Concerning the term "hot-moment" : It was inspired from (McClain et al., 2003 Biogeochemical Hot spots and Hot Moments at the Interface of Terrestrial and Aquatic Ecosystems.): "A hot-moment corresponds as a short period of time with disproportionately high reaction rates relative to longer intervening time periods.". The term is also widely used in the literature. We choose to maintain it in the ms as it conveys our exact message. Concerning the choice of paramount : it has been replaced by "essential" or synonyms. Ex: During the last decade, process-based models have become key tools in estimating carbon and nitrogen budgets in the context of global multi-source changes. Future studies will concentrate in extending the current approach to other tropical basins, needless to say that local observations will be essential for the validation of such exercise and preferably over the same period of analysis.

"- Please have the text proofread by a native English speaker. There are many sentences and phrases which are odd. - There are several (annoying) typos: mole should read mol (various places throughout the text); 'og' should read 'of' in the caption of Fig. 5; N20 should read N2O (Fig. 5); 3rd column/2nd line in Tab. 1: there is something wrong with the exponent; 'anormalies' should read 'anomalies'(P13L15), etc. - Please replace NO3 with NO3- (in the equations as well as throughout the text and figures)"

The manuscript was thoroughly revised to improve the writing and to correct the typos.

[Figure]
Interactive
comment

[Figure]

**Fig. 1.**

---

## Referee Comment (RC2) · Anonymous Referee #2 · 16 Apr 2020

Comment on:

"Denitrification, carbon and nitrogen emissions over the Amazonian wetlands"

Guilhen et al. present a new approach to estimate denitrification rates and associated emissions of the greenhouse gases $CO_2$ and $N_2O$ in Amazonian wetlands. Their method is based on a combination of satellite data, in situ hydrographic monitoring and remote sensing. The study is of high relevance since it aims to use the new method for providing a large-scale assessment of carbon and nitrogen cycling in the Amazonas basin. To date such an endeavor has been limited by the scarcity of available observations. Although I generally think the method used by the authors is sound, the lack of a rigorous explanation of the procedures they followed to setup their model and treat the in situ data does not allow the reader to assess if and how their method can be used to

address the proposed scientific questions. The authors write that their study "aims at delivering an enhanced understanding and quantification of the denitrification process over Amazonian wetlands with their associated fluxes of N2O and CO2". Yet, based on the manuscript it is not clear at what extent the suggested drivers for denitrification in the basin are applicable in situ, and how much of the variability is explained by the model assumptions. Along these lines, I noticed that the authors use wording such as "it is supposed/assumed" several times throughout the manuscript without further substantiating their reasoning. I am aware that any model needs some assumptions; however these need to be based on clear, comprehensible criteria. I believe authors have good reasons to adopt the assumptions they used for their model setup. However, if they are not mentioned in the manuscript, this inevitably affects its credibility. Examples of this are the selection of the N2O/N2 ratio and constant nitrate values over the entire basin during the whole annual cycle. Being nitrate a central parameter for the estimated denitrification rates, this is certainly something that has to be discussed in further detail. Unsubstantiated assumptions are also frequent during the discussion, in which processes or results shown during previous studies are generally considered as valid for the author's investigation without discussing if and what extent they are applicable. In addition to these issues, the manuscript has several flaws in terms of grammar and format consistency. I therefore strongly recommend the authors to revise these aspects on a future version.

All in all, although the manuscript by Guilhen et al. is of scientific significance for environmental sciences, there are several aspects of scientific quality and presentation that need to be addressed before it is considered for publication. Hence, in its present form I do not recommend this manuscript for publication in Biogeosciences. In the following, I list general and specific comments which, in my opinion, could improve the manuscript.

General comments

- Due to the lack of clarity with respect to the model setup and how the denitrification

rates and trace gas emissions were computed, it is hard to grasp whether the specific objectives of the study were successfully tackled or not. In particular, I recommend the authors to clearly differentiate between inferences that apply to natural processes and those which are derived from model-driven improvements and that only tell something about the model's performance.

- After refining the methods (see above and specific comments), I suggest the authors to more clearly formulate their discussion and conclusions in order to highlight the relevance of their study for a wider community (potentially very important in my opinion). At the end of the manuscript I would expect to have answers to the questions: were the drivers for denitrification in the Amazonian wetlands successfully identified? What is the extent of the emissions of CO2 and N2O and how they compare with global estimates?

- There are several misspellings and redaction problems throughout the manuscript, as well as inconsistent presentation of measurement units (several types used for the same variable) and acronyms (abbreviated and spelled in full throughout). I kindly suggest the authors to carry out a careful revision of these aspects to improve the scientific presentation. Also please note that chemical compounds (such as N2O and CO2) should not be italicized.

- Please re-check all figure captions since as they stand they are not informative and rather repeat what is already shown by the figures' legends.

Specific comments

Title: I think CO2 and N2O emissions would be more precise and appropriate for the manuscript.

P.1 l.6 "denitrification and trace gas emissions"

P.1 l.7 "activation-stabilization-deactivation": This is mentioned here and then suddenly during the results. However there is no explanation as to what is the meaning of each

term. Please include a brief description.

P.1 l.14-15 "data driven approach": All studies are data based, please clarify whether you mean data-model-based approach.

P.2 l.4 "(Borges et al., 2015)": This publication deals with CO2 and CH4, not with N2O.

P.2 l.29-30 "However, considering the carbon budget (...)": It seems odd to refer to a sink here when the whole last paragraph is a bout sources. Perhaps this sentence needs to be swapped with the next one.

P.3 l.8 "provide": Perhaps "identify" is more appropriate.

P.4 l.15-16 "Devol et al. (1995)...": This sentence is a repetition of the previous one.

P.4 l.2 "paramount": The word "crucial" would be a better fit here.

P.4 l.10 "floodplain (O-M FP)": replace by "floodplain (in the following O-M FP)".

P.4 l.14 "In situ data and gauging stations data": Please explain clearly which data you are referring to; i.e. which variables you used, their accuracy and spatial resolution.

P.4 l.14-15 Spell all abbreviations in full upon first usage.

P.4 l.16 "with associated quality and uncertainty": What does this mean? It seems the sentence should have ended at "rivers".

P.5 Figure 2: The regional differences are hardly visible with this color bar. Consider replacing it or perhaps using ranges to fewer colors to improve the display. Also, it is a mystery to the reader what V-polarization and 32 means, please clarify.

P.5 Methods: Subsections 2.2 and 2.3 should be moved to this section.

P.5 l.3-9: Please re-check this whole paragraph since it is not clear at all. Also do not italicize chemical compounds. Moreover, I recommend to avoid using the word "suppose" and derivatives since it gives signs of lack of accuracy in your statements.

P.6 l.25-27: If these values were taken from literature, please cite the corresponding sources. Also substantiate your choices and why they are the best for this particular study.

P.6 l-29-31 "On the other hand (. . .) (Sánchez-Pérez et al., 1999)": Rephrase, it would appear as the results on that paper would be results of this study which is of course not the case.

P.7 l.2-3 "Dissolved organic carbon": This was already defined as an abbreviation before in the text. For consistency keep using the abbreviations after first usage.

P.7 l.3 "stable seasonality": Seasonality implies, by definition, changes. I'm guessing the authors mean marked seasonality (i.e. that can be observed reliably every year). Also please state with respect to which parameter this seasonality is strong; is it the discharge?

P.7 l.5 "regarding": Consider replacing by "according to".

P.7 l.5 "main sub-basins": Is this an operational criteria or are there any particularities to the different sub-basins? The authors state that Branco basin has differences with respect to the soil properties and therefore I wonder if and how this would affect your approach of using the same "extrapolation" uniformly.

P.7 l.6 "average monthly discharge": Please show some numbers on this as well as the details of the calculation. How many stations per sub-basin were used? Are there significant differences?

P.7 l.6-7 "We then used those discharge (. . .)": This needs to be explained. Did you use in situ data and extrapolate spatially? If so with which approach? Any caveats that should be considered? Did you grid the data? This is really important since in the current manuscript it comes as a bit of a surprise that the authors present a full map of DOC that sets the basis for some of the large-scale calculations. Without knowing the origin of this data, it is difficult to trust the model results.

P.7 l.7-8 "It was supposed that (. . .)": This seems arbitrary. Is there are reason why it should not change?

P.7 Figure 3 caption: Delete "easily"

P.7 l.11 "nitrates": Please refer to nitrate concentrations instead of nitrates. Alternatively use the chemical formula throughout the manuscript after defining it upon first usage.

P.7 l.16: Consider rephrasing: do you mean "regardless of" instead of "regarding"?

P.8 l.4-5 "Soil data were determined (. . .)": The word "determined" should be replaced by "extracted", "retrieved" or other appropriate option. Also, please specify which parameters were used.

P.8 l.5-6 "The soil description file (. . .)": Consider rephrasing, this sentence is confusing. Also, does this mean that you used in situ nitrate data? If so, this information should have appeared earlier in the manuscript.

P.8 l.7 "nitrogen": Above it says you retrieved nitrate data but here you write that it was derived from the nitrogen contents. Please clarify whether the nitrate values were obtained directly or indirectly. Should the latter be the case, explain how this was done. Also, "contents" is not a precise indication of the magnitude of this variable; please refer to concentrations or other appropriate expression with the corresponding unit.

P.8 l.28-29 "The mean annual denitrification (. . .)": It is not clear to me what is meant with this sentence. Which trends?

P.8. l.29: "Hot moments": In the author's response to the comments of reviewer #1 I saw that this term seems to be widely used in the community and because of this they would prefer to keep it. However the journal has a wide readership and therefore it is appropriate to briefly explain what is meant by this.

P.9 Figure 4: Having a border on the same color as one of the categories of the color bar is not appropriate. Also the image quality does not allow distinguishing the features

described in the text.

P.9 Section 3.2: Reconsider this subtitle since as it stands is not informative as to which is its content.

P.9 l.7: "The following comments can be given:": At this stage I would prefer to talk about results, observations, inferences based on data, or similar, but not "comments". Please consider a different option. Also, using an active voice rather than a passive one would improve the text here and in similar instances. I kindly invite the authors to check for this.

P.9. l.8 "global trend": Probably you mean "overall trend"; using the word "global" here can be misleading because this statement refers to the basin, not the globe.

P.11. Figure 6: Bar plots (e.g. stacked bars) would convey much better the relative contribution of each floodplain to the total denitrification and the emissions of CO2 and N2O.

P.11 l.3: Again, global should be replaced by total, overall or similar in this context.

P.11 l.6-8 "While the O-M floodplain (. . .)": This sentence is confusing. Is the main message here that most of the variability in denitrification and emissions of CO2 and N2O can be explained by the O-M FP and that this result is statistically significant? If so with which level of confidence? What are then the exact values or percentages of the contributions? It is not enough to say that one floodplain is the main source if this is not supported by numbers. I strongly suggest to rephrase and substantiate this statement.

P.12 Figure 7: Change "Denetrification" by "Denitrification" on the y-label axis. Also in this case stacked bars might improve visualization. As for the caption, it contains an unnecessary repetition of information already contained in the plot itself.

P.12 l.2 "are twice as much higher": Based on the numbers in the table I would say the authors mean two orders of magnitude higher rather than twice as much. Please

check.

P.12. l.2 "Averagely": Replace by "In average" or similar.

P.12 Table 1: The exponent notation on the emissions of CO2 and N2 for the Amazon basin should read "x 1010" and not "x 1010". Also, I believe the table captions should go on top of them. Please check the journal's style guidelines.

P.13 l.1-2: "Over the whole basin (. . .)": I am assuming this means no significant trend. If this is correct please state it with a more clear formulation and substantiate with numbers/plots.

P.13 Section 3.2: Replace "gazes emissions" by "trace gas emissions" or "CO2 and N2O emissions".

P.13 l.8-9: "la Niña year": Citation is needed and at best provide an index.

P.13 l.15: Replace "anormalies" by "anomalies" here and subsequent instances. Also, the study covers 2011-2015, not 2010-2015.

P.13 l.16: "were calculated by (. . .)": I suggest rephrasing this sentence. If I understand this correctly, you took the mean of a given month across the years 2011-2015 and then subtract it from each month in the time series to calculate the anomaly. However this does not easily comes across in the text.

P.13 l.18: What is the exact number for "the most"?

P.13. l.19 "However": This word implies contradiction, which in this case does not exist because the second sentence is not related to the first one. Please check.

P.13 l.20 "significant effect on (. . .)": Please state precisely what is meant here. How is an effect measured? Is it the denitrification rate? With significant do you mean a statistically significant difference?

P.13. l.22: "It appears": This does not appear but rather it is exactly what is shown in

Fig. 8. On the other hand, I do not see necessary to plot all years if only 2011 and 2015 are compared. The other years in between only distract the reader. That being said, it is interesting to see that the responses of the Madeira and Branco floodplains are decoupled and completely change sign during La Niña and El Niño years, whereas between 2012 and 2014 they seem to be coupled. A discussion as to why this is the case would argue in favour of keeping the plot as it is.

P.13 l.25-26: "As so, it can be (. . .)": Before it was stated that there were no significant trends; therefore "assuming" here is speculative and contradicts your results.

P.14 l.7: "analysed analytically": Redundant, but beyond that, what does it mean?

P.14 l.9-10 "Overall, the denitrification (. . .)": This sentence is an example of how the variables important for the model and the variables that are key for the processes in situ cannot be clearly distinguished. Please clarify.

P.14 l.14: "processing": It is not clear what this means here.

P.15. l.2: "natural ecosystems": Which ecosystems? References?

P.15. l.7: "sensing waterbodies": Probably here it is meant to say: "(. . .) conducting remote sensing–based monitoring of water bodies".

P.15 l.12 "(equation 1)": Please check the journal's style regulations but I believe here an abbreviation (Eq. 1 or similar) should suffice.

P.15 l-14-15: Spell all abbreviations in full.

P.15 l.19: "kPOC and kDOC": These parameters were taken from the literature. Hence, a reader that is not familiar with the cited work won't understand how temperature, water saturation of the soil, nitrogen contents, soil pH and micro-organisms activity are accounted for. Please clarify.

P.15 l.22-23: How can it be that the N2O emissions from the wetlands only are higher than for the whole basin in which they are included? Please check.

P.15 l.23: "global": See comments above with respect to this term.

P.16 l.1 "We consider it as being produced (...)": This is another statement that is not substantiated at all and leaves open questions as to what the model does. It is crucial for the reader to know this right on the methods section.

P.16 l.12 "simulation node": This is the first time this term appears in the manuscript. Please mention its meaning in the methods section.

P.16 l.15 "critically": Replace by "considerably" or similar.

P.16 l.19 "participate to": Replace by "contribute with".

P.16 l.21-22 "even a small change (...)": This statement is confusing. The authors argue that even small changes could drastically modify the carbon budget. However I would expect this to be supported by a disproportionately high share to the total emissions. Hence, I wonder whether 0.01% is such a high contribution. Should this be the case, I invite the authors to substantiate the statement.

P.16 l.22: "It constitutes (...)": This seems to be a loose sentence here, please check.

P.16. Table 4: Replace "gaz" by "gas". Also, the units can be added to the caption and the last column of the table can be removed.

P.17. l.4: "close": Consider replacing by "similar" / "alike" / "comparable".

P.17 l.19-20 "we may": This expression sounds doubtful and does not reflect confidence in your results. Please consider replacing it.

P.18 l.4 "transpires": This word does not seem correct here. Please check.

P.18. l.6 "(...) depends on rainfalls (...)": Yet, no plot showing discharge is presented.

---

## Author Comment (AC2) · 5 May 2020

Anonymous Referee #2 Comment on: "Denitrification, carbon and nitrogen emissions over the Amazonian wetlands"

Guilhen et al. present a new approach to estimate denitrification rates and associated emissions of the greenhouse gases CO2 and N2O in Amazonian wetlands. Their method is based on a combination of satellite data, in situ hydrographic monitoring and remote sensing. The study is of high relevance since it aims to use the new method for providing a large-scale assessment of carbon and nitrogen cycling in the Amazonas

basin. To date such an endeavour has been limited by the scarcity of available observations. Although I generally think the method used by the authors is sound, the lack of a rigorous explanation of the procedures they followed to setup their model and treat the in situ data does not allow the reader to assess if and how their method can be used to address the proposed scientific questions. The authors write that their study "aims at delivering an enhanced understanding and quantification of the denitrification process over Amazonian wetlands with their associated fluxes of N2O and CO2". Yet, based on the manuscript it is not clear at what extent the suggested drivers for denitrification in the basin are applicable in situ, and how much of the variability is explained by the model assumptions. Along these lines, I noticed that the authors use wording such as "it is supposed/assumed" several times throughout the manuscript without further substantiating their reasoning. I am aware that any model needs some assumptions; however, these need to be based on clear, comprehensible criteria. I believe authors have good reasons to adopt the assumptions they used for their model setup. However, if they are not mentioned in the manuscript, this inevitably affects its credibility. Examples of this are the selection of the N2O/N2 ratio and constant nitrate values over the entire basin during the whole annual cycle. Being nitrate a central parameter for the estimated denitrification rates, this is certainly something that has to be discussed in further detail. Unsubstantiated assumptions are also frequent during the discussion, in which processes or results shown during previous studies are generally considered as valid for the author's investigation without discussing if and what extent they are applicable. In addition to these issues, the manuscript has several flaws in terms of grammar and format consistency. I therefore strongly recommend the authors to revise these aspects on a future version.

All in all, although the manuscript by Guilhen et al. is of scientific significance for environmental sciences, there are several aspects of scientific quality and presentation that need to be addressed before it is considered for publication. Hence, in its present form I do not recommend this manuscript for publication in Biogeosciences. In the following, I list general and specific comments which, in my opinion, could improve the

manuscript.

General comments -

Due to the lack of clarity with respect to the model setup and how the denitrification rates and trace gas emissions were computed, it is hard to grasp whether the specific objectives of the study were successfully tackled or not. In particular, I recommend the authors to clearly differentiate between inferences that apply to natural processes and those which are derived from model-driven improvements and that only tell something about the model's performance. -

After refining the methods (see above and specific comments), I suggest the authors to more clearly formulate their discussion and conclusions in order to highlight the relevance of their study for a wider community (potentially very important in my opinion). At the end of the manuscript I would expect to have answers to the questions: were the drivers for denitrification in the Amazonian wetlands successfully identified? What is the extent of the emissions of CO2 and N2O and how they compare with global estimates? -

There are several misspellings and redaction problems throughout the manuscript, as well as inconsistent presentation of measurement units (several types used for the same variable) and acronyms (abbreviated and spelled in full throughout). I kindly suggest the authors to carry out a careful revision of these aspects to improve the scientific presentation. Also please note that chemical compounds (such as N2O and CO2) should not be italicized.

"We thank the referee for the detailed review of our paper associated with very relevant comments and suggested enhancements. The referee exposed a lack of clarity with respect to the model setup and how the denitrification rates and trace gas emissions were computed. Therefore, we revised parts of the Materials and Methods section to give more clarity on how we built our dataset, our hypothesis and the model performance. We combined sections 2.2 and 2.3 into a new Materials section. Section 2.2

"In situ data from the HyBAm observatory" was improved . We particularly detailed how we built our DOC and NO3 datasets. Descriptions on how we refined the methodology are detailed in the specific comments sections. Our model only simulates denitrification and associate trace gases emissions ($CO_2$ and $N_2O$). In the manuscript, all the consideration made to build our input dataset are now presented. The considerations are linked to the soil texture map database (FAO to assess porosity - nitrogen content in soil (see comment P.8 l.7)) or in situ gauging stations of the HyBAm network (for river discharges and DOC concentrations). Overall, we improved the section 2 "Materials and methods" and the manuscript to detail how we modified the denitrification equation from Peyrard et al., (2010) for the case of the Amazon basin. We clearly separated the assumptions made for building our dataset (DOC and NO3-) and the inferences of the model. The discussion and conclusion sections have been modified to emphasize the inferences brought by our model and to show it successfully answers our stated objectives. According to the model, we identified that both the DOC concentrations and the extent of water bodies (the SWAF values) are the main drivers of the denitrification and trace gases emissions. $CO_2$ emissions from denitrification account for 0.01% of the Amazon Carbon budget and represent a fraction of 3.5 x 10ˆ-6 of the global $CO_2$ emissions (natural and anthropogenic). When we compare our simulated $N_2O$ emissions to other estimations over the Amazon basin we find that our estimations are higher (+ 28%) even though we only take wetlands into account. For that reason, we discussed in the manuscript the importance of distinguishing wetlands in $N_2O$ models as those areas are significant sources of $N_2O$ emissions. The manuscript has been carefully revised to correct all grammatical, misspelling and typo errors. All figures have been regenerated for enhanced presentation and for some better content. We will upload a revised manuscript in the new version as soon as we are invited to do so by the Editors."

————

Please re-check all figure captions since as they stand they are not informative and

rather repeat what is already shown by the figures' legends.

The figure captions were checked and corrected to avoid redundancy with the figure legends. Figures referencing in the text was also enhanced to emphasis on the results.

Figure 1 caption is now: "The Amazon river basin and it's main tributaries mapped over the SRTM digital elevation model."

Figure 2 caption is now: "Monthly averages from 2011 to 2015 of the SWAF surface water fractions over the Amazon basin based on Vertical polarization brightness temperatures (TB V) at 32.5\textdegree{} incidence angle acquired by the SMOS satellite."

Figure 3 caption is now: "Map of the spatial parameters of the denitrification model. DOC contents in mg/L mapped over each sub-basin of the main streams in January 2011 with local observation stations in blue circles (Left). NO3- contents (mol/l) of the watershed over FAO's soil types (Right).

Figure 4 has been updated to enhance visibility and caption is now: "Spatial representation of N2O, CO2 and denitrification summed over the year 2011 to year 2015. The location of the main floodplains (hotspots) are outlined in the Denitrification map."

Figure 5 caption is now: "Monthly denitrification (kg-N), CO {2} (kg-C)and N2O (kg-N) emissions over the entire Amazon watershed for the period 2011 - 2016."

Figure 6 has been updated to include impact of N2O/N2 ratio values showing only N2O emissions and caption is now: "Monthly time series of N2O emissions over the basin, the O-M FP, the Madeira FP and the Branco FP over the period (2011-2016). The lines represent the emissions for a N2O / N2 ratio of 0.1 whereas the coloured areas refer to the potential range of the same ratio : 0.05 - 0.2. Denitrification and CO2 emissions follow the same patterns but with a scale factor of times 10 for denitrification and times 2 for CO2."

Figure 7 has been formatted as a bar plot and caption is now : Average monthly contribution of each floodplain to the Basin denitrification, over the Obidos - Manaus, the

Madeira, Branco floodplains and total denitrification. The residual contribution from the 100\% is associated to the other wetlands in the basin.

Figure 8 caption is now: "Monthly anomalies at the basin and main floodplains scale for denitrification throughout the period (2011-2015)."

Note that table captions where also improved.

P.1 l.6 "denitrification and trace gas emissions"

We added the term "trace gas emissions" in the sentence P1. L.6 which is now:

"Our results show that the denitrification and trace gas emissions present a strong cyclic pattern linked to the inundation processes that can be divided into three distinct phases: activation - stabilization – deactivation"

P.1 l.7 "activation-stabilization-deactivation": This is mentioned here and then suddenly during the results. However, there is no explanation as to what is the meaning of each term. Please include a brief description.

A description was added in the result section 3.1 P.8 l.20. "We find that the denitrification process can be separated into three phases. First the activation phase that is triggered by the increase of the flooded areas and the increase in the microbiological activities. Second, the stabilization phase which corresponds to a maximum denitrification rate and a peak in microbiological activities. And third, the deactivation phase which corresponds to the retreat of inundation which also reduced the microbiological processes of denitrification. Note that this conclusion is not independent of the selected model implementation and associated assumptions."

Also, in the limitation section of the discussion the following paragraph was added on potential enhancement that can be done to better simulate the processes. P17 L27

" Considering the dynamics of the activation-stabilization-deactivation of the denitrification, they can be more precisely assessed if variables like water surface temperatures

and water depth were added in the future. These variables can inform on the speed at which the activation and deactivation of the microbiological process of denitrification are triggered."

P.1 l.14-15 "data driven approach": All studies are data based, please clarify whether you mean data-model-based approach.

"Data driven approach" was used to identify that the model implementation and calibration is based on data information and numerical approaches. In their book's (Hydrological data driven approach) Introduction, https://doi.org/10.1007/978-3-319-09235-5 (Remesan & Mathew, 2015) explain that They "explore a new realm in data-based modelling with applications to hydrology." So, we are in phase with the reviewer comment as data driven can be considered as a sub-approach to data-based-modelling. And as the term data-based-modelling can be considered as more universal then "data driven approach" it was replaced in the text by "data-base methodology".

P.2 l.4 "(Borges et al., 2015)": This publication deals with CO2 and CH4, not with N2O. An additional reference for N2O was added. The sentence is now: "This phenomenon is intimately linked to nitrous oxide (N2O) (Wu et al., 2009) and carbon dioxide (CO2) (Borges et al., 2015) emissions to the atmosphere."

Wu, J., Zhang, J. and, J. W., Xie, H., Gu, R., Li, C., and Gao, B.: Impact of COD/N ratio on nitrous oxide emis-sion from microcosm wetlands and their performance in removing nitrogen from wastewater, Bioresource Technology, 100,30https://doi.org/https://doi.org/10.1016/j.biortech.2009.01.056, 2009.

P.2 l.29-30 "However, considering the carbon budget (. . .)": It seems odd to refer to a sink here when the whole last paragraph is about sources. Perhaps this sentence needs to be swapped with the next one.

The two sentences were combined and are now:

"In regards to the carbon budget, some studies show that the Amazon basin is more

or less in balance and even acts as a small sink of carbon at the amount of 1GtC/yr (Lloyd et al., 2007)."

P.3 l.8 "provide": Perhaps "identify" is more appropriate. "Provide" was replaced by "to identify". The sentence is now: "The specific objectives of the study are to highlight the main key factors controlling the denitrification and to identify the hot-spots and hot-moments of denitrification over wetlands."

P.4 l.15-16 "Devol et al. (1995). . .": This sentence is a repetition of the previous one.

The previous sentence P.3 l.13-14: "The Amazon hydrology is governed by three main sources: the Andes, the Brazilian and Guyana shields and the lowlands. " was removed.

P.4 l.2 "paramount": The word "crucial" would be a better fit here.

The three uses of paramount were replaced in the manuscript: P2 L11: replaced by "key tools" P4 L2: replaced by "essential" In conclusion: replaced by "essential"

P.4 l.10 "floodplain (O-M FP)": replace by "floodplain (in the following O-M FP)".

Modified, the sentence is now:

"Here we consider the three main floodplains: the Branco floodplain in the northern part, the Madeira floodplain in the southern part and the floodplain between Odidos and Manaus which is called Obidos - Manaus floodplain (in the following O-M FP)"

For the sake of homogeneity, all through the manuscript text the Branco Floodplain, Madeira floodplain and Obidos-Manaus Floodplain are refered to as Branco FP, Madeira FP and O-M FP.

A rigorous check on the use of acronyms was also done.

P.4 l.14 "In situ data and gauging stations data": Please explain clearly which data you are referring to; i.e. which variables you used, their accuracy and spatial resolution.

The paragraph (section 2.2) concerning the used datasets was clarified as follow:

"In situ data were obtained from the HyBAm long-term monitoring network that maintains, in collaboration with the national stakeholders and local universities, 13 gauging stations in the Amazon catchment basin since 2003. For the Brazilian part of the basin, a network of eight local stations is maintained by the French Research Institute for Development (IRD) and the Amazonas Federal University (UFAM). Geochemical, sedimentary and hydrological data are available freely at www.so-hybam.org for each of the gauging stations. River discharge records are available daily while geochemical data, including DOC, are available monthly. In our study we extracted both the daily river discharges and the monthly DOC concentrations. The list of stations we used in the study are found in Fig. 3 (left)."

P.4 l.14-15 Spell all abbreviations in full upon first usage. Abbreviations were tracked throughout the manuscript and corrected to ensure uniformity. For SMOS, Hybam, POC, DOC,

P.4 l.16 "with associated quality and uncertainty": What does this mean? It seems the sentence should have ended at "rivers".

The sentence was updated as in the above answer (P4L14). " P.5 Figure 2: The regional differences are hardly visible with this color bar. Consider replacing it or perhaps using ranges to fewer colors to improve the display. Also, it is a mystery to the reader what V-polarization and 32 means, please clarify.

The figure caption was updated as stated above and an explanation was added to in the text to explain V-polarization: "The SMOS satellite observes the Earth surface at full polarization (Horizontal (H) and Vertical (V) polarization) with multi-incidence-angles variable incidence angles." More detailed information is found in Parrens et al., (2017) for the description of the SWAF product and (Al Bitar et al. 2017) for the brightness temperature angle binned brightness temperature products from SMOS.

Fig. 2 was reworked using a different set of colours to improve the visibility.

Figure 2: Monthly averages from 2011 to 2015 of the SWAF surface water fractions over the Amazon basin based on Vertical polarization brightness temperatures (TB V) at 32.5° incidence angle acquired by the SMOS satellite." P.5 Methods: Subsections 2.2 and 2.3 should be moved to this section.

A separate Materials sub-section was added after subsection: study site in which the two paragraphs were added. Now Sub-section "2. Materials and Methods" contains a Materials subsection divided into: 2.2 Materials 2.2.1 In situ data from the HyBAm observatory 2.2.2 Water surface extents from L-Band microwave

P.5 l.3-9: Please re-check this whole paragraph since it is not clear at all.

We changed to paragraph P.5 l.3-9 to improve the clarity of our methodology. The paragraph is now:

"In this study, we modified the denitrification rate proposed by Peyrard et al. (2010) to fit tropical wetland conditions. Denitrification is the consumption of dissolved organic carbon (DOC), particulate organic carbon (POC) and nitrate ($NO_3$-) in the soil. This process is limited by dioxygen ($O_2$) and ammonium ($NH_4$+) availability. Denitrification occurs during flood events when the soil has low oxygen concentrations, thus $O_2$ concentration is not a limiting factor (Dodla et al., 2008). Furthermore, as there is only one long flood pulse in the Amazon watershed, we consider that all the ammonium is processed into nitrate between two consecutive floods. We also consider that $NH_4^+$ is not a limiting factor. The fact that nitrate stocks are reconstituted by nitrification under aerobic conditions, e.g when soils are no longer flooded, is a reasonable assumption in the case of the Amazon basin and more particularly for the wetland parts as shown by (Brettar et al., 2002) upper Rhine floodplain. Besides, many studies consider denitrification as a combined consumption of nitrates and carbon (Scofield et al. (2016); Dodla et al. (2008); Goldman et al. (2017)). Taking into consideration the above statements, the denitrification rate is expressed as:"

Also do not italicize chemical compounds. All chemical compounds are now in non-italic font throughout the manuscript. The "chemformula" package was used for the chemical equations.

Moreover, I recommend to avoid using the word "suppose" and derivatives since it gives signs of lack of accuracy in your statements. The word "suppose" was replaced in the manuscript by "consider". P.6 l.25-27: If these values were taken from literature, please cite the corresponding sources. Also substantiate your choices and why they are the best for this particular study.

Concerning porosity, the following descriptions were added to the text: "Soil Porosity ($\varphi$) is computed based on the soil texture from the FAO database at 11 km resolution and other studies (Sun et al., 2015). The porosity is averaged over the computation nodes (25x25km) using a bilinear interpolation." Concerning kPOC, kDOC, kNO3: "kPOC, kDOC, kNO3 are obtained from Sun et al., (2015) who performed a study of denitrification over the Garonne catchment (temperature anthropogenic watershed). We adapted these parameters to the case of the Amazon basin. The parameters result from the best simulation. To our knowledge these parameters were never measured over the Amazon basin and the value we used are the best published estimates that we have."

P.6 l-29-31 "On the other hand (. . .) (Sánchez-Pérez et al., 1999)": Rephrase, it would appear as the results on that paper would be results of this study which is of course not the case.

The sentence was changed to: "On the other hand, Sánchez-Pérez et al., (1999) showed that when denitrification is active during flooding event, nitrate pool of wetlands is provided and sustained by nitrate content coming from streams, in the case of the forested Rhine floodplain."

P.7 l.2-3 "Dissolved organic carbon": This was already defined as an abbreviation before in the text. For consistency keep using the abbreviations after first usage.

As mentioned above all abbreviations are now well mentioned in the manuscript.

P.7 l.3 "stable seasonality": Seasonality implies, by definition, changes. I'm guessing the authors mean marked seasonality (i.e. that can be observed reliably every year). Also please state with respect to which parameter this seasonality is strong; is it the discharge?

The sentence is clarified as follows: "In terms of discharge, the marked seasonality of the Amazonian streams was demonstrated by prior studies (...)"

P.7 l.5 "regarding": Consider replacing by "according to". The paragraph was rewritten. Similar misuse of the word "regarding" were checked throughout the manuscript.

P.7 l.5 "main sub-basins": Is this an operational criteria or are there any particularities to the different sub-basins? The authors state that Branco basin has differences with respect to the soil properties and therefore I wonder if and how this would affect your approach of using the same "extrapolation" uniformly.

The delineation applied in the study is based on the gauging stations used in the study. The location of the stations is in Fig.3 left below

Figure 3: Map of the spatial parameters of the denitrification model. DOC contents in mg/L mapped over each sub-basin of the main streams in January 2011 with local observation stations in blue circles (Left). $NO_3^-$ contents (mol/l) of the watershed over FAO's soil types (Right).

Therefore, we spatialized our data (DOC) from the gauging station to the sub-basin. The figure was replotted to better show the separation between the sub-basins. Soil property is one particularity that is considered in the computation of the $NO_3^-$ dataset. $NO_3^-$ concentrations are associated with a given type of soil. The impact of soil type on Nitrate concentration is shown in fig.3 right. In his study, Ludwig et al., (1996) demonstrated that DOC concentration was a function of discharge, soil carbon content and slope. In our study, we analysed the DOC measurements over the HyBAm stations

and noticed the marked seasonality. We then used discharge to estimate spatialized DOC concentrations. Thus, this methodology draws a general trend of DOC behaviour with average values. As so, it narrows the variability of DOC concentrations for one sub basin.

P.7 l.6 "average monthly discharge": Please show some numbers on this as well as the details of the calculation. How many stations per sub-basin were used? Are there significant differences?

Fig.3 (left) above, now displays the gauging stations used for the study. We selected the stations considering few criteria: availability of DOC and discharge data location of the stations. Some stations are out of the flooded areas (shown by the SWAF data), so we decided to discard them for the study as we only consider DOC in the flood-plains. Overall, we have one station per main sub basin. For the average monthly discharge, we extracted the mean discharge of each month from each station. Finally, we calculated the mean average discharge for each month on the basis of daily measurements. In order to clarify our calculation, we changed "average monthly discharge" to "the mean monthly discharge" in the section 2.4.2 (2.3.2 in the new manuscript)

P.7 l.6-7 "We then used those discharge (. . .)": This needs to be explained. Did you use in situ data and extrapolate spatially? If so with which approach? Any caveats that should be considered? Did you grid the data? This is really important since in the current manuscript it comes as a bit of a surprise that the authors present a full map of DOC that sets the basis for some of the large-scale calculations. Without knowing the origin of this data, it is difficult to trust the model results.

The DOC data is computed on the basis of each sub-basin using the relation between DOC and discharge provided in Ludwig et al. (1996). We then associated the calculated values to the main sub basin. For example, for the Madeira sub-basin (see new Fig.3 left) we selected only one gauging station. Discharge and DOC were extracted from the station. We then built our DOC dataset using our methodology based on

the hydrology marked seasonality and the relationship between DOC and discharge. Eventually, the DOC values were extended to the whole Madeira sub basin. The paragraph is now modified as follows: "The daily discharge was extracted from the gauging stations used in the study (Fig. 3} (left)) from the HyBAm database (1983 – 2012). For each station, we calculated the mean monthly discharge from the daily observations. In terms of discharge, the marked seasonality of the Amazonian streams was demonstrated by prior studies (Paiva et al., 2013). For the DOC concentrations, we extracted the monthly measurements for the same stations over the same period. As the SWAF's periods (2011 – 2015) and the DOC measurements are not concomitant, we calculated a mean average monthly DOC concentration for each station. When the information of DOC concentration was not available, our dataset was gap filled using a linear relationship between DOC concentration and discharge (Ludwig et al., 1996) based on the discharge marked seasonality of the Amazonian streams. Finally, we extended the calculated values to the associated main sub basin."

Ludwig, W., Probst, J.-L., and Kempe, S.: Predicting the oceanic input of organic carbon by continental erosion, Global Biogeochemical Cycles, 10, 23–41, https://doi.org/10.1029/95GB02925, http://onlinelibrary.wiley.com/doi/10.1029/95GB02925/abstract, 1996.

P.7 l.7-8 "It was supposed that (. . .)": This seems arbitrary. Is there are reason why it should not change?

In the paragraph, we state the marked seasonality in terms of discharge for the streams. Moreover, DOC concentration is linked to discharge (DOC = f(Q)) (Ludwig et al., 1996) so DOC has the same property as discharge and shows a marked seasonality. Hence, DOC concentrations change little from year to year (as discharge). Thus we considered that the monthly time series of DOC (for each sub-basin) are similar for the different years.

Overall, section 2.4.2 (now 2.3.2 in the manuscript) was changed to: "The model

parameters for the denitrification are taken from references studies and in situ measurements. The sediment porosity $\varphi$ was set to 25%. It is computed based on the soil texture from the FAO database at 11 km resolution and other studies (Sun et al., 2015). The porosity is averaged over the computation nodes (25x25km) using a bilinear interpolation kP OC , kDOC and kNO3 were calibrated to $1.6 \times 10-7$ d $-1$ , $8 \times 10-3$ d $-1$ and 30 $\mu$molL$-1$ respectively. They are obtained from Sun et al., (2015) who performed a study of denitrification over the Garonne catchment (temperate anthropogenic watershed). We adapted these parameters to the case of the Amazon basin. The parameters result from the best simulation. To our knowledge these parameters were never measured over the Amazon basin and the value we used are the best published estimates that we have. For P OC concentration, according to the studies performed by Moreira-Turcq et al. (2013), it was considered constant over the whole watershed and for the global period of the simulation (2011 – 2015) to 10 %. The daily discharge was extracted from the gauging stations used in the study (Fig. 3 (left)) from the HyBAm database (1983 – 2012). For each station, we calculated the mean average discharge for each month on the basis of daily measurements. In terms of discharge, the marked seasonality of the Amazonian streams was demonstrated by prior studies (Paiva et al., 2013). For the DOC concentrations, we extracted the monthly measurements for the same stations over the same period. As the SWAF's periods (2011 – 2015) and the DOC measurements are not concomitant, we calculated a mean average monthly DOC concentration for each station. When the information of DOC concentration was not available, our dataset was gap filled using a linear relationship between DOC concentration and discharge (Ludwig et al., 1996), based on the discharge marked seasonality of the Amazonian streams. Finally, we extended the calculated values to the associated main sub basin.

Nitrate concentrations were calculated for every type of soil given by the FAO's classification in the upper 30 cm layer. Batjes and Dijkshoorn (1999) drew a complete description of total the nitrogen content of the soils of the Amazon region. Evaluating nitrates in the upper layer of the soils was executed adapting the mineralization rate

which is based on the average temperature of the region and the proportion of both clay and limestone. For the most biologically active soils, as gleysols and fluvisols, the mineralization rate was set up to 7% of the organic nitrogen amount, which is the maximum observed value in the region. On the contrary, regosols are biologically less active soils with mineralization rates hardly reaching 2% (Legros, 2007; Sumner, 1999). Finally, we determined the nitrate concentrations by combining the nitrate content in each type of soil with the water storage capacity for each type of soil (L), deducted from the FAO soil database. Nitrate concentrations (NO3-) were considered constant over the period. On the one hand, as the Amazon is one of the most active region of the world (Legros, 2007) in term of microbial soil dynamic, it was assumed that on the one hand, during non-flooding period, mineralization of nitrogen was sufficient to compensate nitrate loses by plant assimilation and leaching. On the other hand, Sánchez-Pérez et al., (1999) showed that when denitrification is active during flooding event, nitrate pool of wetlands is provided and sustained by nitrate content coming from streams, in the case of the forested Rhine floodplain."

P.7 Figure 3 caption: Delete "easily" A new caption is provided (see answer to first comment).

P.7 l.11 "nitrates": Please refer to nitrate concentrations instead of nitrates. Alternatively use the chemical formula throughout the manuscript after defining it upon first usage.

The chemical formula NO3- is now used in the manuscript instead of nitrate. It was applied to all chemical compounds such as NH4+, O2, etc.

P.7 l.16: Consider rephrasing: do you mean "regardless of" instead of "regarding"?

The sentence P7 l.16: "We consider that the gases produced during the denitrification are entirely emitted to the atmosphere regarding the supersaturation of pCO2 in groundwater (Davidson et al., 2010)" Was changed to: "Because of the supersaturation of pCO2 in groundwater (Davidson et al., 2010), we consider that CO2 and N2O produced during denitrification are entirely emitted to the atmosphere."

P.8 l.4-5 "Soil data were determined (. . .)": The word "determined" should be replaced by "extracted", "retrieved" or other appropriate option. Also, please specify which parameters were used. The word determined was replaced by "retrieved'. We referred to the NO3- concentrations that were calculated for each type of soils. We added a paragraph on that statement; see comments below.

P.8 l.5-6 "The soil description file (. . .)": Consider rephrasing, this sentence is confusing. Also, does this mean that you used in situ nitrate data? If so, this information should have appeared earlier in the manuscript. The sentence is removed and all description of the use of soil types is now assembled in the previous section. We did not use in situ data to build our NO3- dataset as most of NO3- measurements are performed in streams and we focus on floodplain soils.

P.8 l.7 "nitrogen": Above it says you retrieved nitrate data but here you write that it was derived from the nitrogen contents. Please clarify whether the nitrate values were obtained directly or indirectly. Should the latter be the case, explain how this was done. Also, "contents" is not a precise indication of the magnitude of this variable; please refer to concentrations or other appropriate expression with the corresponding unit.

To clarify our methodology we added in section 2.4.2 a short paragraph on how we generated our nitrate dataset: "Nitrates were calculated for every type of soil given by the FAO's classification in the upper 30 cm layer. Batjes and Dijkshoorn (1999) drew a complete description of total the nitrogen content of the soils of the Amazon region. Evaluating nitrates in the upper layer of the soils was executed by correcting the mineralization rate for a given soil. For the most biologically active soils, as gleysols and fluvisols, the mineralization rate was set up to 7% of the organic nitrogen. On the contrary, regosols are biologically less active soils with mineralization rates hardly reaching 2% (Legros, 2007; Sumner, 1999). Finally, we determined the nitrate concentrations by combining the nitrate content in each type of soil with the water storage capacity for each type of soil (L), deducted from the FAO soil database."

From P8 l.3 to P8 l.11: the paragraph was reworked to : "where DNO3 is the net denitrification in mol/month, RNO3 is the denitrification rate in mol/month/L, SWAF is the fraction of land covered with open waters and Qwa is the water storage capacity for each type of soil (L), deducted from the FAO soil database. In summary the model requires the inputs and parameters for : (1) the nitrate concentration for each type of soil (mol/L), (2) the DOC concentrations of the streams that overflow, extended to the associated sub-basin and (3) the extent of inundated surfaces. The model was applied at monthly scale from January 1st 2011 to December 31th 2015 and monthly maps were then generated. Note that in order to assess the denitrification only occurring in wetlands, the minimum SWAF value recorded during the period (2011 - 2015) is subtracted to each month simulation, as it accounts as a residual artefact of streams."

P.8 l.28-29 "The mean annual denitrification (. . .)": It is not clear to me what is meant with this sentence. Which trends? Here we refer to the average general trend of denitrification observed over one year for the whole basin (e.g Fig. 7 black line). It shows the three different phases (activation - stabilization - deactivation). For more clarity, we added few lines at the beginning of the paragraph:

"The average monthly denitrification over the basin for the period 2011 - 2015 (depicted in Fig. 7 as the black line) represents the main trend observed over the Amazonian watershed. We find that the denitrification process can be separated into three phases activation – stabilization - deactivation. First the activation phase that is triggered by the increase of the flooded areas and the increase in the microbiological activities. Second, the stabilization phase which corresponds to a maximum denitrification rate and a reach of microbiological activities And third, the deactivation phase which corresponds to the retreat of inundation which also reduced the microbiological processes of denitrification."

P.8. l.29: "Hot moments": In the author's response to the comments of reviewer #1 I saw that this term seems to be widely used in the community and because of this they would prefer to keep it. However the journal has a wide readership and therefore it is

appropriate to briefly explain what is meant by this.

The definition and the reference for the expression "hot moment" were added in the paragraph P.8 l.29 before detailing them: "A hot-moment corresponds to a short period of time with disproportionately high reaction rates relative to longer intervening time periods (McClain et al., 2003)."

P.9 Figure 4: Having a border on the same color as one of the categories of the color bar is not appropriate. Also the image quality does not allow distinguishing the features described in the text.

The figure was changed to avoid confusions with the different colours and to improve the quality of the image.

Figure 4: Spatial representation of N2O, CO2 and denitrification summed over the year 2011 to year 2015. The location of the main floodplains (hotspots) are outlined in the Denitrification map.

P.9 Section 3.2: Reconsider this subtitle since as it stands is not informative as to which is its content.

Changed to : "Denitrification, CO2 and N2O emissions: focus on the three main Amazon floodplains"

P.9 l.7: "The following comments can be given:": At this stage I would prefer to talk about results, observations, inferences based on data, or similar, but not "comments". Please consider a different option. Also, using an active voice rather than a passive one would improve the text here and in similar instances. I kindly invite the authors to check for this.

The sentence was replaced by "the results of the model provide the following inferences:". Moreover, we corrected the manuscript to avoid the use of the passive way.

P.9. l.8 "global trend": Probably you mean "overall trend"; using the word "global" here

can be misleading because this statement refers to the basin, not the globe.

The word "global" was replaced by "overall" when relevant.

P.11. Figure 6: Bar plots (e.g. stacked bars) would convey much better the relative contribution of each floodplain to the total denitrification and the emissions of $CO_2$ and $N_2O$.

We understand that the reviewer is mentioning figure 7 bar plots, we answer the question related to it in the corresponding comment. Concerning figure 6 it has also been updated to include the impact of the physical extent values of the NO2/N ratio.. Also onlyN2O emissions are shown now.

P.11 l.3: Again, global should be replaced by total, overall or similar in this context.

Answered above.

P.11 l.6-8 "While the O-M floodplain (. . .)": This sentence is confusing. Is the main message here that most of the variability in denitrification and emissions of $CO_2$ and $N_2O$ can be explained by the O-M FP and that this result is statistically significant? If so with which level of confidence? What are then the exact values or percentages of the contributions? It is not enough to say that one floodplain is the main source if this is not supported by numbers. I strongly suggest to rephrase and substantiate this statement.

In this section we tried to determine if the three floodplains have a different or similar contribution to the denitrification / emissions. To do so, we run an ANalysis Of VAriance (ANOVA) on our complete time series (monthly value from 2011 to 2015) with a level of confidence alpha = 5%. The results indicate that the FP have indeed an impact on the processes (p.value = $1.35 \times 10^{-8}$). We then ranked the three FP by applying a post-hoc analysis (same alpha and p.value). The analysis revealed 2 separated groups: group A = the O-M FP, group B = the Madeira FP and the Branco FP. On the one hand, we found that the O-M FP is the main source of emissions; on average it provides 38% of the

emissions. On the other hand, we found that the Branco and the Madeira floodplains contribute similarly to the processes (on average 25% and 21% respectively).

In the manuscript P.11 l.6-8 was changed to : "We then run an ANOVA and a post-hoc analysis to determine the contribution to the basin denitrification of each floodplain. The results return two different groups (p.value = 1.35 x 10ˆ-8, alpha = 5%). The first group is constituted by the O-M FP which is the main source of denitrification for the basin and provides 38% of the process on average. The second group is constituted by the Branco and the Madeira floodplains. They contribute similarly to the processes (on average 25% and 21% respectively)."

P.12 Figure 7: Change "Denetrification" by "Denitrification" on the y-label axis. Also in this case stacked bars might improve visualization. As for the caption, it contains an unnecessary repetition of information already contained in the plot itself.

The y-label was corrected to "Denitrification". Fig.7 was replotted as stacked bars with an updated caption.

Figure 7: Average monthly contribution of each floodplain to the Basin denitrification, over the Obidos - Manaus, the Madeira, Branco floodplains and total denitrification. The residual contribution from the 100\% is associated to the other wetlands in the basin.

P.12 l.2 "are twice as much higher": Based on the numbers in the table I would say the authors mean two orders of magnitude higher rather than twice as much. Please check. Corrected in Review 1.

P.12. l.2 "Averagely": Replace by "In average" or similar. The correction was made and replaced by "on average". No other use of "averagely" was found in the manuscript.

P.12 Table 1: The exponent notation on the emissions of $CO_2$ and $N_2$ for the Amazon basin should read "x 1010" and not "x 1010". Also, I believe the table captions should go on top of them. Please check the journal's style guidelines.

Exponents were corrected.

P.13 l.1-2: "Over the whole basin (. . .)": I am assuming this means no significant trend. If this is correct please state it with a more clear formulation and substantiate with numbers/plots.

Indeed, we observed no significant differences of yearly emissions at watershed scale during the period for both $CO_2$ and $N_2O$ emissions. In the manuscprit we changed from P.12 l1.3 to P.13 l.1-2 to : "Table 1 depicts the yearly emissions of $CO_2$ and $N_2O$ over the Amazon basin and the three main floodplains. Emissions of $CO_2$ from denitrification are twice as much higher than $N_2O$ emissions over the basin. The total yearly emissions of $CO_2$ and $N_2O$ over the Amazon basin are significantly identical from 2011 to 2015 (Kruskal-Wallis p.value = 0.9929). On average, flooded areas produce 2.76 × 109 kg C-$CO_2$ per year and 1.03 × 109 kg N-$N_2O$ per year by denitrification from the natural $NO_3^-$ pool of the watershed." P.13 Section 3.2: Replace "gazes emissions" by "trace gas emissions" or "$CO_2$ and $N_2O$ emissions".

The title was change to "Denitrification and trace gas emissions anomalies"

P.13 l.8-9: "la Niña year": Citation is needed and at best provide an index.

A reference was added that details el Nino and la Nina events on the Amazon basin. "On the one hand, year 2011 was a "la Niña year" (Moura et al. 2019).

New ref: https://doi.org/10.1016/j.scitotenv.2018.09.242

P.13 l.15: Replace "anormalies" by "anomalies" here and subsequent instances. Also, the study covers 2011-2015, not 2010-2015.

Modified.

P.13 l.16: "were calculated by (. . .)": I suggest rephrasing this sentence. If I understand this correctly, you took the mean of a given month across the years 2011-2015 and then subtract it from each month in the time series to calculate the anomaly. However this

does not easily comes across in the text.

The calculation explanation was changed P13 l15 -16 to: "Anomalies were determined by first calculating the mean value for each month across the period 2011 - 2015. This mean value was then subtracted from each corresponding average month in the series."

P.13 l.18: What is the exact number for "the most"?

We substantiated our inferences with more results in the manuscript P.13 l.18: "(...) during La Niña year and the heavy precipitations period, most of the anomalies are positive especially for the first months (66% - 66% for the total denitrification, 16% - 83% for the O-M FP, 25% - 33% for the Madeira FP and 100% - 50% for the Branco FP respectively)."

P.13. l.19 "However": This word implies contradiction, which in this case does not exist because the second sentence is not related to the first one. Please check.

Agreed. The sentence P.13 l.19 starts now as : "During el Nino episode (. . .)"

P.13 l.20 "significant effect on (. . .)": Please state precisely what is meant here. How is an effect measured? Is it the denitrification rate? With significant do you mean a statistically significant difference?

Here we performed an ANOVA on our time series (monthly values from 2011 to 2015) to assess if a meteorological event had an impact on the processes. We associated each month to an event; el Nino, la Nina, heavy rainfall or regular (as factor). The results show that only El Nino has an effect on the denitrification/emissions. To evaluate this difference, we compared the value simulated during the months el Nino to the average value of this given month. For example, Dec 2015 was under el Nino conditions. So, we compared the simulated value of Dec 2015 to the mean value of all the Dec during 2011 - 2015. On average we find that during el Nino conditions denitrification / emissions are 27% lower than the expected mean values.

P.13. l.22: "It appears": This does not appear but rather it is exactly what is shown in Fig. 8. On the other hand, I do not see necessary to plot all years if only 2011 and 2015 are compared. The other years in between only distract the reader. That being said, it is interesting to see that the responses of the Madeira and Branco floodplains are decoupled and completely change sign during La Niña and El Niño years, whereas between 2012 and 2014 they seem to be coupled. A discussion as to why this is the case would argue in favour of keeping the plot as it is.

The sentence P.13 l.22 now starts as: "extreme meteorological events do not have an uniform impact on the whole basin."

We included all the years in our analysis (ANOVA). Moreover, heavy precipitations were recorded from October 2013 to March 2014. We think that it is important to notice that la Nina and heavy rainfall lead to wetter conditions but have no impact on the denitrification and the emissions. Two conclusions are made here. First, only el Nino event has an impact on the processes; all the anomalies are all negative. Second, the other events (la Nina and heavy precipitations) have no impact on the processes. Moreover, no clear general trends for the anomalies were found during those events. The referee's comments are correct and interesting nevertheless we believe that no conclusion can be made regarding the fact that only one la Nina and el Nino events are recorded over the study time span.

P.13 l.25-26: "As so, it can be (. . .)": Before it was stated that there were no significant trends; therefore "assuming" here is speculative and contradicts your results.

(Table 2) We indeed found no significant trend at a yearly time step for the whole basin in general. When we focus on the three floodplains: the emissions of the OM FP and the Madeira FP increase but with a small statistical significance whereas for the Branco FP it decreases and denitrification is reduced by a factor 2 from 2011 to 2015. We associated the drying of the Branco FP as a reason of the decrease of its denitrification. This is consistent with our model design, the decrease is attributed to

the decrease of the inundated surfaces of the Branco FP.

we revised the manuscript accordinglyP.13 l.24-26 "The average denitrification rate for the whole basin shows little inter annual variations. However, in 2015 simulated denitrification for the Branco FP was twice as low than for the year 2011. As so, it can be assumed that this floodplain has been drying off during the 2011 - 2015 period and thus is much more sensitive to drier conditions than other parts of the watershed"

Was modified to:

"The average yearly denitrification rates for the whole basin, the O-M FP and the Madeira FP show no clear trend between 2011 and 2015. For the Branco FP, a decreasing trend was identified during the study period. From 2011 to 2015 the simulated average yearly denitrification for the Branco FP drops by a factor two."

P.14 l.7: "analysed analytically": Redundant, but beyond that, what does it mean?

Corrected, that was a redundant mistake. We meant analysed

P.14 l.9-10 "Overall, the denitrification (. . .)": This sentence is an example of how the variables important for the model and the variables that are key for the processes in situ cannot be clearly distinguished. Please clarify.

We refer here to the variables that are important to the model. We further clarified our statement by changing P.14 l.10 to: "DOC and SWAF are the main driving variables of the denitrification model." P.14 l.14: "processing": It is not clear what this means here.

Here we wanted to emphasize on the biogeochemical processing potential of wetlands. To clarify our statement. In the manuscript we completed the sentence P.14 l.14 as :

"The denitrification values show that all the three floodplains are particularly active systems in terms of ecological services for processing organic matter and NO3-. "

P.15. l.2: "natural ecosystems": Which ecosystems? References?

This statement was a loose sentence. The section P.15 l1-2 "The Branco floodplain, which is the bottom value of the set with an average potential of 38.8 kgN/ha/yr, has values at least twice as much higher than natural ecosystems." was removed from the text as it confuses the reader and it is not of interest considering the paragraph.

P.15. l.7: "sensing waterbodies": Probably here it is meant to say: "(. . .) conducting remote sensing–based monitoring of water bodies".

The sentence was changed to: "This result strengthens the importance of Earth Observation (EO) based monitoring of water bodies for determining inundated surfaces patterns and intensities and their impact on biochemical processes."

P.15 l.12 "(equation 1)": Please check the journal's style regulations but I believe here an abbreviation (Eq. 1 or similar) should suffice.

All equations are references as Eq. X in the updated manuscript.

P.15 l-14-15: Spell all abbreviations in full.

The model's abbreviations are now spelled in full: N2O Model Inter-comparison Project (NMIP) Dynamic Land Ecosystem Model (DLEM) Vegetation Integrative SImulator for Trace gases (VISIT) Organising Carbon and Hydrology In Dynamic Ecosystems - Carbon Nitrogen (ORCHIDEE-CN)

P.15 l.19: "kPOC and kDOC": These parameters were taken from the literature. Hence, a reader that is not familiar with the cited work won't understand how temperature, water saturation of the soil, nitrogen contents, soil pH and micro-organisms activity are accounted for. Please clarify.

We added a brief description to explicitly explain how temperature and others are accounted for in kPOC and kDOC. "kPOC and kDOC are the mineralization rate parameters. They describe the kinetic processing of organic matter into POC and DOC respectively. The organic matter processing is performed by microbial communities. Therefore, environmental conditions such as temperature and soil pH have a direct

influence on the bacterial activity and turnover. The cumulated impact of temperature, soil pH and micro-organisms activity, is accounted indirectly for in our approach through the parameters kPOC and kDOC and the mineralisation rate describe in Eq. 1 ((Peyrard et al., 2010; 20 Sun et al., 2017))."

P.15 l.22-23: How can it be that the N2O emissions from the wetlands only are higher than for the whole basin in which they are included? Please check.

The results are correct. The difference is due to the spatial extent that is considered. There is one total amount of N2O for the whole basin that originates from the flooded areas. But it can be weighted by the basin area or the wetlands area. So we have 2 different values of weighted emissions: the wetland is higher because associated to the smaller area.

P.15 l.23: "global": See comments above with respect to this term. Answered above

P.16 l.1 "We consider it as being produced (. . .)": This is another statement that is not substantiated at all and leaves open questions as to what the model does. It is crucial for the reader to know this right on the methods section.

This consideration is made when building the NO3 database. Our model only simulates denitrification along with CO2 and N2O emissions. The new section 2.3.2 (2.4.2 in the non revised ms) is clearer on how we built our NO3 dataset and details the considerations with relevant references (see answers above) and removes any confusion.

P.16 l.12 "simulation node": This is the first time this term appears in the manuscript. Please mention its meaning in the methods section. For clarification, we refer to the computation over nodes and not pixels as we use the EASEv2 25km grid, which has rectangular nodes that conserve surface but not dimensions over latitudes. We changed P.8 l.11: "The model was applied at daily scale from January 1st 2011 to December 31th 2015 and monthly maps were then generated." to: "The model simulations were applied over the EASEv2 nodes at daily scale from January 1st 2011 to

December 31th 2015 and monthly maps were then generated. "

P.16 l.15 "critically": Replace by "considerably" or similar.

We changed P.14 l.15: "Our wetlands estimations are critically lower (10ˆ4) than integrated ecosystem observations." To: "Our wetlands estimations are considerably lower (10ˆ4) than integrated ecosystem observations." No other use of the word "critically" was found in the text

P.16 l.19 "participate to": Replace by "contribute with". The sentence P.16 l.19: "Overall, CO2 emissions from denitrification over the whole Amazon basin participate to 0.01% of the carbon emissions of the watershed." Was change to: "Overall, CO2 emissions from denitrification over the whole Amazon basin contribute with 0.01% of the carbon emissions of the watershed." No other use of the word "participate" was found in the text

P.16 l.21-22 "even a small change (. . .)": This statement is confusing. The authors argue that even small changes could drastically modify the carbon budget. However I would expect this to be supported by a disproportionately high share to the total emissions. Hence, I wonder whether 0.01% is such a high contribution. Should this be the case, I invite the authors to substantiate the statement.

Actually, the comment confirms our statement, but we need to clarify it. Our point is that wetlands contributes little in terms of emissions so if the wetland area is converted to, lets say, crop land or managed forests this change will drastically impact the emissions budget as the crop land for example will have a higher contribution. Thus, a small change in the natural wetlands cover to an anthropogenic non-inundated area has a big impact on the emissions budget. This change can also be due to climatic events like dry El-Nino events. In order to reflect our statement we rephrased the sentence. The following sentence:

"Most of the CO2 emissions over the Amazon are attributed to processes such as organic matter respiration from biomass. Confirming previous studies, this result means that even a small change in the distribution of wetlands cover over the Amazonian basin may drastically modify the carbon budget." Was changed to: "Most of the CO2 emissions over the Amazon are attributed to processes such as organic matter respiration from biomass and little contributions from wetlands. Previous study from Vicari et al. 2011 showed that the change of wetlands into forested area can increase the carbon emissions drastically. In this context and in light of the results obtained in this paper one can conclude that in case of very dry natural events or intense anthorpogenisation of the land-cover the carbon budget of the once wetland areas and now non-inundated surfaces will greatly increase." Vicari, R., Kandus, P., Pratolongo, P., and Burghi, M.: Carbon budget alteration due to landcover-landuse change in wetlands: the case of afforestation in the Lower Delta of the Parana River marshes (Argentina), WATER AND ENVIRONMENT JOURNAL, 25, 378–386, https://doi.org/10.1111/j.1747-6593.2010.00233.x, 2011.

P.16 l.22: "It constitutes (. . .)": This seems to be a loose sentence here, please check.

We believe that regarding the current context (land use change / deforestation) over the Amazon basin, it's important to highlight practices that impact C and N balance. The sentence was modified to: "This constitutes an important topical subject for the Amazonian basin."

P.16. Table 4: Replace "gaz" by "gas". Also, the units can be added to the caption and the last column of the table can be removed.

"Gaz" was replaced by "gas". The last column of the table was removed and the units were moved to the legend.

P.17. l.4: "close": Consider replacing by "similar" / "alike" / "comparable". We changed P.17 l.4 "The N2O emissions from the Amazon and the Congo basins are close" To: "The N2O emissions from the Amazon and the Congo basins are comparable"

We also changed the term "close" in P.8 l.26 : "Values registered in September are lower than in August, and yet in year 2011, 2012 and 2015, these were similar." P.1 l.12-13 : "(. . .) we found that the Amazonian wetlands have similar emissions of N2O to the tropical Congo wetlands" P18 l.6: "Overall, the results appear alike to other large scale models; especially for N2O emissions."

P.17 l.19-20 "we may": This expression sounds doubtful and does not reflect confidence in your results. Please consider replacing it.

In the revised manuscript we corrected all the sentence that sounded doubtful and rephrased them when necessary.

P.17 l19-20: Moreover, our results show that the Òbidos - Manaus floodplain possesses the same denitrification potential as a nitrate polluted temperate ecosystem." P.13 l.23 "Overall denitrification may not be impacted at watershed scale" Changed to: "Extreme meteorological events do not impact the denitrification and trace gases emissions at the basin scale." P14 l.8-9 "NO3- is a non-limiting factor for the denitrification In the Amazon basin". P.15 l.11-12 "Overall, the denitrification rate (Eq. 1) shall be considered as a combination of a potential rate function (provided by DOC and POC) and limitation functions provided by the peculiar environmental conditions." P.18 l.5-6: "Each floodplain possesses its own functioning that depends on rainfalls and the hydrology of the floodplain's river."

P.18 l.4 "transpires": This word does not seem correct here. Please check.

We understand the concerns of the Referee on using the correct words. According to the Oxford Dictionary, "transpire" means "Prove to be the case / Occur, Happen" which reflects the wanted message.

P.18. l.6 "(. . .) depends on rainfalls (. . .)": Yet, no plot showing discharge is presented. We added in section 4.5 "Limitations of the current approach" a plot that shows the discharge at Obidos as well as the simulated denitrification. It shows that denitrification is

[Figure]

triggered few weeks before the flooding and is maximum during that period. It indicates that precipitations and flooding both have a key role in the denitrification process.

[Figure]

[Figure]

**Fig. 1.** Monthly averages from 2011 to 2015 of the SWAF surface water fractions over the Amazon basin based on Vertical polarization brightness temperatures (TB V) at 32.5° incidence angle acquired by the SMOS

[Figure]

**Fig. 2.** Spatial representation of N2O, CO2 and denitrification summed over the year 2011 to year 2015. The location of the main floodplains (hotspots) are outlined in the Denitrification map.

**Fig. 3.** Monthly averages from 2011 to 2015 of the SWAF surface water fractions over the Amazon basin based on Vertical polarization brightness temperatures (TB V) at 32.5° incidence angle acquired by the SMOS

---

## Referee Report (RR1)

**Comment on:**

**"Denitrification, carbon and nitrogen emissions over the Amazonian wetlands"**

In an earlier review of this manuscript I expressed the potential relevance of the study albeit my concerns with respect to the lack of clarity on the approach used to reach the conclusions postulated by the authors. Having read the revised version submitted by the authors, it is my opinion that the issues raised (both of content and format) were satisfactorily addressed. Hence, after addressing the technical corrections listed below, I do recommend this manuscript for publication in Biogeosciences.

**Technical corrections**

Note that these corrections are referred to the file named:

"bg-2020-3-author_response-version1.pdf"

P.1 l.3 "from the denitrification": delete "the"

P.2 l.41 (...) pointed out that over the Amazonian wetlands disproportionally (...)

P.3 l.59 (...) a priori in situ information (...)

P.6 l.113 Replace "dioxygen" by "oxygen"

P.7 l.144 "(25x25km)":  Add space between the numbers and the unit of measure.

P.7 l.148 Replace "For PCO" by "The" and delete "it" from the sentence "(…) it was considered (…)"

P.11 l.221 "Denitrification as well as the $CO_2$ and $N_2O$ emissions (...)"

P.11 l.248 "We then ran": Delete "then"

P.12 l.257 Replace "p.value" by "p-value"

P.16 l.309 Replace "facts" by "fact" and "assesses" by "determines" or similar.

P.16 l.310 Replace "(…) floodplains. Whereas (…)" by "floodplains, whereas".

P.16 l.314 Replace "are compared" by "were compared".

P.16 l.322 "accounted for indirectly"

P.17 l.349-350 Consider replacing "Previous study from (…)" by "The study by Viocari et al. (2011) (...)".

P.19 l.389 Delete ":"

P.19 l.397 "Carbon" and "Nitrogen" should be written as "carbon" and "nitrogen" here

P.19 l.403 "(…) emphasize the importance of (…)"

P.19 l.406 Replace ")" by a comma.

---

## Author Response (AR2)

**Point-by-point response**

**Review 1**

Comment on:

"Denitrification, carbon and nitrogen emissions over the Amazonian wetlands"

In an earlier review of this manuscript I expressed the potential relevance of the study albeit my concerns with respect to the lack of clarity on the approach used to reach the conclusions postulated by the authors. Having read the revised version submitted by the authors, it is my opinion that the issues raised (both of content and format) were satisfactorily addressed. Hence, after addressing the technical corrections listed below, I do recommend this manuscript for publication in Biogeosciences.

**Technical corrections**

Note that these corrections are referred to the file named:

"bg-2020-3-author_response-version1.pdf"

P.1 l.3 "from the denitrification": delete "the"

**The sentence P.1 L.3 "In this paper, we quantify the CO2 and N2O emissions from the denitrification over the Amazonian wetlands." was changed to:**

**"In this paper, we quantify the CO2 and N2O emissions from denitrification over the Amazonian wetlands"**

P.2 l.41 (...) pointed out that over the Amazonian wetlands disproportionally (...)

**The sentence P.2 L.41 "Scofield et al. (2016) pointed out over the Amazonian wetlands that the disproportionally high CO2 out-gassing may be explained by the amount of podzols for the Negro Basin" was changed to:**

**"Scofield et al. (2016) pointed out that over the Amazonian wetlands disproportionally high CO2 out-gassing may be explained by the amount of podzols for the Negro Basin"**

P.3 l.59 (...) a priori in situ information (...)

**The sentence P.3 L.59 "We constrained and adapted a denitrification process-based set of equations by L-Band microwave water surface extents from the Soil Moisture and Ocean Salinity (SMOS) satellite and a priori information from in situ" was modified to:**

**"We constrained and adapted a denitrification process-based set of equations by L-Band microwave water surface extents from the Soil Moisture and Ocean Salinity (SMOS) satellite and a priori in situ information"**

P.6 l.113 Replace "dioxygen" by "oxygen"

**The sentence P.6 L.113 was changed to "This process is limited by oxygen (O2) and ammonium (NH4+) availability**

P.7 l.144 "(25x25km)": Add space between the numbers and the unit of measure.

**P.7 L.144 the unit of measure was replaced to (25 km x 25 km)**

P.7 l.148 Replace "For PCO" by "The" and delete "it" from the sentence "(…) it was considered (…)"

**The sentence P.7 L.148 "For POC concentration, according to the studies performed by Moreira-Turcq et al . (2013), it was considered constant over the whole watershed and for the entire period of the simulation (2011-2015) to 10%" was changed to:**

**"According to the studies performed by Moreira-Turcq et al . (2013), the POC concentration was considered constant over the whole watershed and for the entire period of the simulation (2011-2015) to 10%"**

P.11 l.221 "Denitrification as well as the CO2 and N2O emissions (…)"

**The sentence P.11 L.221 "The Denitrification, the CO2 and N2O emissions follow the pattern" was changed to:**

**"Denitrification as well as the CO2 and N2O emissions follow the same pattern"**

P.11 l.248 "We then ran": Delete "then"

**P.11 L.248 "We ran an Analysis of Variance (ANOVA) and a post-hoc analysis (…)"**

P.12 l.257 Replace "p.value" by "p-value"

**p.value was checked and replaced by p-value throughout the manuscript.**

P.16 l.309 Replace "facts" by "fact" and "assesses" by "determines" or similar.

**Corrected**

P.16 l.310 Replace "(…) floodplains. Whereas (…)" by "floodplains, whereas".

**The sentence P.16 L.309-310 "As a matter of facts, DOC assesses the average maximum denitrification rate of a floodplain. Whereas the SWAF value is the main driven factor of the model which reveals the actual denitrification" were changed to:**

**"As a matter of fact, DOC determines the average maximum denitrification rate of a floodplain, whereas the SWAF value is the main driven factor of the model which reveals the actual denitrification"**

P.16 l.314 Replace "are compared" by "were compared".

**The sentence P.16 L.314 "The N2O emissions at large scale are compared to the results of N2O models (…)" was changed to:**

**"The N2O emissions at large scale were compared to the results of N2O models (…)"**

P.16 l.322 "accounted for indirectly"

**P.16 L.322 "The cumulated impact of temperature, soil pH and microorganisms activity, is accounted indirectly for in our approach (…)" was changed to:**

**"The cumulated impact of temperature, soil pH and microorganisms activity is accounted for indirectly in our approach (…)"**

P.17 l.349-350 Consider replacing "Previous study from (…)" by "The study by Viocari et al. (2011) (…)".

**P.17 L.349-350 the sentence now starts as "Vicary at al. (2011) showed that (…)"**

P.19 l.389 Delete ":"

**Corrected**

P.19 l.406 Replace ")" by a comma.

**P.19 L389 & P.19 L.406 Typo mistakes were checked throughout the manuscript and corrected. No other occurrences were found.**

P.19 l.397 "Carbon" and "Nitrogen" should be written as "carbon" and "nitrogen" here

**P.19 L.397 "The study also contributes to better understand the functioning of the major floodplains of the Amazon Basin and their respective involvement in the Amazon Carbon and Nitrogen budget" was replaced by:**

**"The study also contributes to better understand the functioning of the major floodplains of the Amazon Basin and their respective involvement in the Amazon carbon and nitrogen budget"**

P.19 l.403 "(…) emphasize the importance of (…)"

**P.19 L.403 "We emphasize on the importance of (…)" was changed to "We emphasize the importance of (…)"**

**Review 2**

Comments on 'Denitrification and associated nitrous oxide and carbon dioxide emissions from the Amazonian wetlands,

ms by Guilhen et al. re-submitted to Biogeosci Discuss; ms# bg-2020-3.

The manuscript (ms) under review is significantly revised compared to the previous version. Although I note that most of my and rev#2's comments have been addressed, I still have severe concerns about the presented approach of estimating the NO3- loss and the associated emissions. In brief, I think it is not reasonable to ignore DNRA and nitrification. Moreover, there are too many typos and the English writing still needs a critical revision by a native speaker. Therefore, I cannot recommend publication of the ms in its present form.

**We understand that the reviewer has comments on the non-inclusion of the DNRA and nitrification processes and we provide here solid justifications for not adding them during flood periods in the wetlands of the Amazon basin and added in the discussion the need to considering DNRA in other ecosystems.**

General Comments

By narrowing their view on denitrification, the authors are missing the opportunity to get a comprehensive idea about the major processes and parameters relevant for N2O an CO2 emissions during the flooding events. There is no word on potentially missing processes and the resulting associated uncertainties in the discussion of the shortcomings of their approach in section 4.5 'Limitation of the current approach'.

**We kindly remind the reviewer that the aim of our study is to focus on heterotrophic denitrification that is the only processes involved in N2 production (Hu et al., 2015) and to assess the part of denitrification in N2O and CO2 emissions at large scale over flooded areas. While only denitrification is considered, efforts have been invested in providing a methodology to compute it at the whole Amazonian basin scale using remote sensing datasets.**

**During the review process additional information were added on the impact of uncertainties in the parameters (figure 6) to answer the reviewer's comments. Also, we have in the discussion section a specific paragraph on the limitation of the study with respect to the stated objectives.**

**Nevertheless, for future more comprehensive studies, the discussion section is now extended to the issue of DNRA and the need to consider it in other environments than the one considered in this study when the approach is extrapolated to other regions. Also, we added a comment on the opportunities to complexify the proposed methodology by integrating additional physical processes (DNRA, nitrification) in order to consider non-flooded periods and other ecosystems.**

**Hu et al., (2015) (https://doi.org/10.1093/femsre/fuv021)**

Specific comments:

- I would like to point out again, that DNRA can be a significant (anoxic) loss process for NO3-. When looking at Table 1 in Rutting et al. (BG, 2011) I find that 12-50% of the overall NO3- loss can be attributed to DNRA, e.g. in a Brazilian forest lowland. I agree that N2O emissions from DNRA might be as low as 1%. However, when assuming that a significant fraction of the NO3- loss is happening via the denitrification channel but instead via the DNRA channel, the total N2O emissions will be reduced significantly as well. So, you may need to modify your approach (equations 1-4) and include DNRA.

In the current study, we quantify denitrification during flood events. We showed in section (4.1) that NO3- in soil solution is a non-limiting factor for the process. In this case, denitrification and DNRA are not competing on the NO3- pool.

Rutting et al. (2011) states in the conclusion that: "we conclude that DNRA is a significant, or even dominant, NO3 consumption process in some ecosystems (Table 1)." And the list of the ecosystems related to this statement can be identified in Table 1. This statement is then followed by: « As Burgin and Hamilton (2007) concluded for aquatic systems, more work is also needed to understand the importance of DNRA in various terrestrial ecosystems.". Our study addresses terrestrial aquatic ecosystems as we provide denitrification during flood events in wetlands.  Thus, we conclude that in the current status of knowledge there is no strong scientific justification to take into consideration DNRA in our conditions. Also, if we consider the DNRA it would be very difficult to justify the parameter values to control it.  Again, while in Rutting et al. (2011) it is stated that: "However, under NO3- limiting and strongly reducing conditions, a shortage of electron acceptors is most likely limiting microbial growth. Under these conditions DNRA has the advantage over denitrification since more electrons can be transferred per mole NO3- (Tiedje et al., 1982)". There is no mention for the NO3- non-limiting conditions with high microbial activity.

Moreover the result of Rutting et al., (2011) shows the estimations of the DNRA rate for Brazilian lowland from Sotta et al., (2008). In the latter study, the authors performed a local measurement in the Caxiuana, National Forest. The study concerned two types of soils (sandy – clay Oxisol). And measurements were performed during wet and dry seasons but not during flooding. We made a google earth snap shot of the location.

[Figure]

Considering all of the elements brought above, we added the following text to the discussion section:

P.19 L.390 : "Sixth, denitrification and dissimilatory nitrate reduction to ammonium (DNRA) are two natural processes for NO3- reduction. In their, review Rutting et al. (2011) states that DNRA competition for NO3- should be considered for some ecosystems which did not include aquatic ecosystems. They added that more studies are needed for terrestrial aquatic ecosystems based on Burgin and Hamilton (2007). Tiedje et al., (1982) showed that under NO3- limiting and strongly reducing conditions, DNRA has the advantage over denitrification.  Sotta et al., (2008) estimated at 12-50% the reduction of NO3- from DNRA in low land Brazilian forest but in non-flooded periods. In

our case, NO3- is non-limiting, thus we do not need to take into account the impact of NO3- loss from DNRA. Moreover, since estimates of the DNRA direct contribution to N2O emissions is about 1% (Cole et al., 1988) and considering the uncertainty and errors linked to the modelling of denitrification in the wetlands of the whole Amazon basin the DNRA processes were not considered. Finally, in our study we focused on denitrification solemnly. In order to provide a complete nitrogen budget for the whole Amazon basin, future studies will need to complexify the proposed methodology by integrating additional biogeochemical processes (DNRA, nitrification,…) and physically relative datasets (soil temperature, soil moisture,…) in order to extend the approach to non-flooded periods and other ecosystems. ”

Rutting et al. (2011) (10.5194/bg-8-1779-2011)

Burgin and Hamilton (2007) (https://doi.org/10.1890/1540-9295(2007)5[89:HWOTRO]2.0.CO;2)

Tiedje et al., (1982) (https://doi.org/10.1007/BF00399542, 1982.)

Sotta et al., (2008) (https://doi.org/10.1016/j.soilbio.2007.10.009)

Cole et al., (1988) (Assimilatory and dissimilatory reduction of nitrate to ammonia: in The Nitrogen and Sulphur Cycles)

- I still wondering why nitrification as a source of N2O under low O2 during the flooding events is ignored. Nitrification does not take place in anoxic conditions (here I agree with the authors) but it can occur along the O2 gradients in the overlying water or along the O2 gradients the upper soils. N2O emissions from nitrification are enhanced when O2 concentrations are approaching anoxic conditions. This will lead to an additional source of N2O. This additional N2O source via nitrification should be mentioned or even included in Eq 1-4. The missing N2O from nitrification may be one of the reasons for the discrepancy between the model results and the measurements (see Table 4). I see that this line of argumentation in contrast to the authors' assumption that nitrification is not taking place during the flooding events, see P6L115-118: 'The fact that NO3- stocks are reconstituted by nitrification under aerobic conditions, e.g. when soils are no longer flooded, is a reasonable assumption in the case of the Amazon basin and more particularly for the wetland parts as shown by (Brettar et al., 2002) on the upper Rhine flood plain.' however, I do not think that this is a reasonable assumption, because the overlying water is in contact with atmosphere, so there is a O2 gradient towards the soil surface (or in shallow well-mixed waters in the upper soil) allowing nitrification to take place. Moreover, Brettar et al. (2002) is not a suitable reference to justify the authors' statement because it deals only with denitrification in flooded soils and nitrification is not mentioned in the publication at all.

We agree with the statements of the reviewer on nitrification, but we kindly remind the reviewer that the aim of our study is to focus on N2O and CO2 emissions due to denitrification under flood conditions. We added in the discussion the potential interest of apply such methodology in future studies that combines parsimonious physical modelling to remote sensing to address other processes like nitrification.

For the missing N2O please see next comment.

On our statement: *'The fact that NO3- stocks are reconstituted by nitrification under aerobic conditions, e.g. when soils are no longer flooded, is a reasonable assumption in the case of the Amazon basin and more particularly for the wetland parts'.* In inundated areas in the Amazon Basin, the intensity of the flood that can last few weeks supports the fact that soils are in anoxic conditions.

The study of Brettar at al. (2002) gives information on the dynamic of the redox potential in soil which is a proxy of nitrification / heterotrophic denitrification. In another work in the groundwater of the alluvial floodplain of the Garonne river, Iribar et al., 2015 showed that denitrification is the principal way to produce N2 via heterotrophic denitrification and quantify the NosZ genes involved in the heterotrophic denitrification. We understand the worries of the reviewer therefore, for a complete description of the nitrogen dynamic in the Upper Rhine forest, the reference in the manuscript was changed to Sánchez-Pérez and Trémolières (2003)

In the case of assessing the part of denitrification, DNRA and nitrification on N2O emissions, we agreed with the reviewer on the fact that measurements on the bacteria communities should be performed to assess the part of denitrification/nitrification/DNRA to improve those kinds of study.

Further answers concerning the N2O emissions are provided in the next answer.

Sánchez-Pérez and Trémolières (2003) (https://doi.org/10.1016/S0022-1694(02)00293-7)

Iribar et al., (2015) (https://doi.org/10.1016/j.ecoleng.2015.02.002)

- An important reference about N2O emissions has been overlooked: Liengaard et al., Hot moments of N2O transformation and emission in tropical soils from the Pantanal and the Amazon (Brazil); Soil Biol. Biochem., 75, 26-36, 2014.

The study presented by Lienggard et al., (2014) aims at detailing the production of N2O during hot moments of N2O emission from both Pantanal and Amazon wetland soils following controlled waterlogging of the soil matrix. We included a reference to this study in section 4.3 'Wetlands and integrated ecosystem emissions'.

We extracted the average simulated values of N2O emissions (from 2011 to 2015) from the closest simulation nodes (25 x 25 km) to the in-situ measurements of Lienggard et al. (2014). Comparing our simulations to the study of Liengaard et al., (2014) draws the same results (section 4.3 P.17 L.342; e.g our simulations are lower of a factor 1/100). But looking closely to Liengaard et al., (2014) the measurement of the N2O emission are occurring "in the best conditions" (e.g maximum inundation, optimal temperature, etc). To reproduce these "best conditions" we looked for a simulation node with maximum flooding. The highest pixel in our simulation is located in the O-M FP for August 2015.

We find that this specific value is in range with the average measurements of Liengaard et al., (2014) (4.9×107 gN/km²/yr while our highest simulation value estimated an emission of about 2.6 ± 1.3 x 107 gN/km²/yr."). This comforts us with our previous comments and the same conclusions (section 4.5) when comparing to local in-situ data that over a 25 x 25 km area the amount of flooded area from monthly data is much lesser than the representative flooded area of the hourly in-situ data.

P.17 L.340 was changed to :

"We extracted the average simulated value of the period from the simulation node. When comparing the N2O with in situ campaigns performed by Koschorreck (2005), Keller et al. (2005) and Liengaard et al. (2014) at the different locations, the wetlands emissions from our study are roughly lower from a factor 10+2 of the integrated ecosystem observed emissions. This difference comes from different spatial and temporal scales for both the in situ measurements and our model. To verify this explanation, we extracted the maximal pixel value simulated during the period of the study. On average, in situ measurements return emissions of about 4.9×107 gN/km²/yr while our highest simulation value estimated an emission of about 2.6 ± 1.3 x 107 gN/km²/yr."

Minor comments

P2L27: What do you mean with '… alternations between terrestrial and aquatic phases in wetlands …'? Do you mean the dry and wet seasons?

**P.2 L.27 was changed to "(…) alternations between dry and wet periods in wetlands (…)"**

P2L43: What do you mean with ' … is more or less in balance and even acts as a small sink …' ? Is it a sink or not?

**Current knowledge on the CO2 budget of the Amazon Basin is that it is more or less in balance considering the uncertainties in the estimates. Some studies consider that it acts as a small sink.**

**We consider the statement clear and we also add that answering the question about the total carbon balance of the Amazonian basin while of utter importance is not the objective of our paper.**

Lloyd, J., Kolle, O., Fritsch, H., De Freitas, S. R., Silva Dias, M. A. F., Artaxo, P., Nobre, A. D., De Araojo, A. C., Kruijt, B., Sogacheva, L., Fisch, G., Thielmann, A., Kuhn, U., and Andreae, M. O.: An airborne regional carbon balance for Central Amazonia, Biogeosciences, 4, 759–768, https://hal.archives-ouvertes.fr/hal-00297719, 2007.

P3L79: Fig 3 is referenced before Fig 2

P9L204: Fig 7 is referenced before Fig 6

**Following the modifications during the review process the referencing was impacted. The Figures appear now in order in the text.**

P6L125: typo 'refres'

**P.6 L.125 typo corrected to refers**

P7L140: 'Nevertheless, with no precise field measurements an average N2O/N2 ratio of 0.1 (Weieretal.,1992) applied in the study.' I guess 'was' is missing before 'applied'.

**P.7 L.140 is now "Nevertheless, with no precise field measurements an average N2O/N2 of 0.1 (Weier et al.,1992) was applied in the study."**

P7L147/148: ' … are the best published estimates that we have …' How do you know that these are the best, because they are the only ones? (which is not a good argument)

**P.7 L.147 was changed to "… are the only published estimates that we have"**

P17L352: What do you mean with 'anthropogenisation'? I do not think that this term is commonly used and I do not understand it in this context.

**P.17 L.352 The term "anthropogenisation" does exist. Still we replaced it with a more commonly used term: "anthropogenic changes" to mention changes induced by human activities.**

P17L349: 'Previous study …' It must read 'A previous study … 'or 'The previous study …'.

**P.17 L.349 changed to "Vicari et al., (2011) showed that (…)"**

P18L370: It must read 'North Sea'

P18L385: 'equifinalities'? What do you mean with this? I do not think that this term is commonly used and I do not understand it in this context. According to Wikipedia 'equifinality' means: The principle that in open systems a given end state can be reached by many potential means, viz. "All roads lead to Rome," or "There's more than one way to skin a cat."

**Actually, equifinality is a very commonly used term to mention compensation effects in parameter identification: (e.g. the same solution (flux values of N2) can be obtained with several sets of parameter values (surface of inundated area / rate of micro-biological activity).**

**For example this highly cited paper by Prof. Been K. in hydrological modelling.**

**Beven, K. (2006). A manifesto for the equifinality thesis. Journal of hydrology, 320(1-2), 18-36.**

**Also the term is commonly used in other papers**

**"equifinality" and "geochemistry":**

**https://scholar.google.fr/scholar?hl=fr&as_sdt=0,5&q=equifinality+geochemistry**

**"equifinality" and "hydrology":**

**https://scholar.google.fr/scholar?hl=fr&as_sdt=0%2C5&q=equifinality+hydrology&btnG=**

P19L397: It must read 'carbon and nitrogen'.

**Typos were corrected and checked throughout the ms.**

.

[revised manuscript text omitted]

---

## Author Response (AR3)

**Relevant changes made in the manuscript**

[Figure]

*Figure 3 Map of the spatial inputs of the denitrification model. DOC contents in mg/L mapped over each sub-basin of the main streams (January) with local observation gauging stations in blue circles (Left). The Amazon watershed is divided into 8 major sub-basins: (1) the Negro basin, (2) the Branco basin, (3) the Solimoes River and its tributaries, (4) the Madeira basin, (5) the Purus basin, (6) the Tapajos basin, (7) the Xingu basin and (8) the section between Manaus and the mouth of the Amazon River. NO3- contents (mol/L) of the watershed over FAO's types of soils (Right).*

[Figure]

*Figure 4 Spatial representation of $N_2O$ emissions (kgN-$N_2O$/km²), denitrification (mol of $NO_3$) and $CO_2$ emissions (kgC-$CO_2$/km²) summed over the year 2013. The locations of the main floodplains (hot spots) are outlined in the denitrification map.*

[Figure]

*Figure 7 Average monthly contribution of each floodplain: the O-M FP (black), the Madeira FP (grey), Branco FP (white) to the Amazon total denitrification. The residual contribution from the 100% is associated with the other wetlands in the basin. The blue line represents the average monthly denitrification for the period of the study and it shows the main trend observed over the Amazon watershed.*

[Figure]

*Figure 8 Monthly anomalies at the basin and main floodplains scale for denitrification throughout the period (2011-2015).*

The writing was revised and the English was corrected; all changes are marked-up in the following manuscript.

[revised manuscript text omitted]